# The evolutionary history and genomics of European blackcap migration

Kira Delmore[1][†]*, Juan Carlos Illera[2], Javier Pérez-Tris[3], Gernot Segelbacher[4], Juan S Lugo Ramos[1], Gillian Durieux[1], Jun Ishigohoka[1], Miriam Liedvogel[1]*

[1]Behavioural Genomics, Max Planck Institute for Evolutionary Biology, Plön, Germany; [2]Research Unit of Biodiversity (UO-CSIC-PA), Oviedo University, Mieres, Spain; [3]Department of Biodiversity, Ecology and Evolution, Complutense University of Madrid, Madrid, Spain; [4]Wildlife Ecology and Management, University Freiburg, Freiburg, Germany

**Abstract** Seasonal migration is a taxonomically widespread behaviour that integrates across many traits. The European blackcap exhibits enormous variation in migration and is renowned for research on its evolution and genetic basis. We assembled a reference genome for blackcaps and obtained whole genome resequencing data from individuals across its breeding range. Analyses of population structure and demography suggested divergence began ~30,000 ya, with evidence for one admixture event between migrant and resident continent birds ~5000 ya. The propensity to migrate, orientation and distance of migration all map to a small number of genomic regions that do not overlap with results from other species, suggesting that there are multiple ways to generate variation in migration. Strongly associated single nucleotide polymorphisms (SNPs) were located in regulatory regions of candidate genes that may serve as major regulators of the migratory syndrome. Evidence for selection on shared variation was documented, providing a mechanism by which rapid changes may evolve.

*For correspondence:
delmore@evolbio.mpg.de (KD);
liedvogel@evolbio.mpg.de (ML)

Present address: [†]Department of Biology, Texas A&M University, College Station, United States

**Competing interests:** The authors declare that no competing interests exist.

## Introduction

Bird migration is a fascinating and highly variable behaviour that integrates many traits – morphological, physiological and behavioural. Research on a wide range of species has provided important insight into this behaviour, from the incredible distances that birds cover during their journeys (*Alerstam et al., 2003*; *Egevang et al., 2010*), to the fine-tuned and precisely controlled timing of migration (*Gwinner and Helm, 2003*) and the fascinating sensory modalities that allow birds to navigate with amazing precision (*Mouritsen, 2018*; *Wiltschko and Wiltschko, 1972*). The European blackcap *Sylvia atricapilla* is an iconic migratory species that is well suited to work on the genetics of migration. Blackcaps exhibit dramatic differences in migratory behaviour (*Figure 1a*), spanning the entire spectrum from exclusively migratory populations in the northern portion of their range to short distance and partially migratory populations in the Mediterranean; non-migratory, or resident, populations occur on both the European continent (Iberian Peninsula) and the Atlantic islands. In addition to variation in the propensity to migrate and the distance covered, blackcaps vary in migratory orientation, with a migratory divide (contact zone between populations that breed adjacent to one another but take different migratory routes) occurring between populations that migrate southwest (SW) and southeast (SE) from their breeding grounds in Central Europe in the autumn. A novel migratory route also evolved very recently in this species, with an increasing number of birds migrating northwest (NW) from the Central European breeding grounds in the autumn (*Cramp, 1992*).

Variation in the migratory behaviour of European blackcaps was harnessed in a series of influential papers published in the 1980s and 1990s that detailed the genetic basis of migration. Common garden experiments showed that selectively bred individuals that were reared in isolation from their

**eLife digest** Every year as the seasons change, thousands of animals migrate huge distances in search of food or better climates. As far as migrations go, there might be none so impressive as the trans-oceanic flights made by small migrating songbirds. These birds can weigh as little as three grams and travel up to 15,000 kilometres. Most migrate alone and at night and yet still manage to return to the same location each year. Several strands of research suggest there could be a genetic basis to their migratory behaviour, but exactly which genes control this phenomenon remains poorly understood.

One small songbird that has been studied for decades is the European blackcap. This species exhibits a real variety of migration patterns. Some blackcaps travel rather short distances, others much further, and some populations do not migrate at all. Populations that share the same breeding grounds in the summer may migrate in different directions in the autumn. These features make it a good species to study the genetic variation between populations that migrate in different directions and over different distances. However, only in recent years has advancing technology made it possible to comprehensively study an animal's entire genome, leaving no gene unturned.

Now, Delmore et al. have used high-throughput sequencing technologies to trace the evolutionary history of migration in European blackcap and started by assembling a reference genome for the species. Then, the genomes of 110 blackcaps from several populations that take different annual migrations were compared to the reference. This revealed that the populations began to diverge some 30,000 years ago and that there was some apparent gene mixing between groups of migrating and resident blackcaps around 5,000 years ago. The analysis showed only a small set of genes code for their differences in migration. Additionally, while the candidate genes were shown to be common among blackcaps, the genes identified did not match those reported from studies of other migrating songbirds. Finally, Delmore et al. also noted that the differences between the populations tend to be in the parts of the genome that control whether a given gene is switched on or off, which could explain how new migratory behaviours can rapidly evolve.

This study is one of the most comprehensive genomic analysis of migration to date. It is important work as songbirds, like other animals, are responding to increasing pressures of environmental and climate change. In time, the findings could be used to support conservation efforts whereby genetic analyses could determine if certain populations possess enough variation to respond to coming changes in their habitats.

parents maintain population-specific behaviour, suggesting that there is a genetic component to migration (*Helbig, 1994*; *Helbig et al., 1989*; *Helbig, 1991a*; *Berthold and Querner, 1981*; *Pulido and Berthold, 2010*). F$_1$ hybrids crossbred between populations that differ in migratory traits exhibited intermediate phenotypes (orientation, distance and propensity to migrate), suggesting that these traits are additively inherited (*Berthold and Querner, 1981*; *Pulido and Berthold, 2010*; *Helbig, 1991b*). Further work with F$_2$ hybrids showed a wider distribution of phenotypes and the recovery of parental phenotypes, indicative of traits that are controlled by only a few major genes (*Helbig, 1996*), and selection experiments mating birds according to migratory status showed that the transition between resident and migratory behaviour can occur in just a few generations (*Pulido et al., 1996*; *Berthold et al., 1990*). The rapidity with which migratory behaviour can evolve has been supported in natural populations; the NW route taken by some birds was only established in the past 70 years and probably in response to increased food availability during the winter in the United Kingdom (*Berthold et al., 1992*).

The blackcap has also been the subject of extensive phylogeographic study. *Pérez-Tris et al. (2004)* used mitochondrial (control) data from 241 birds and 12 populations across the entire breeding range to show that migratory variation in this species arose recently (4,000–13,000 years ago [ya]) and has not yet resulted in significant population differentiation. These results could suggest that the genes that control migratory variation have small effect sizes or are restricted to a small portion of the genome. The only populations showing substantial genetic differentiation occurred in the Central European migratory divide (i.e., between SW and SE migrants), indicating that differences in orientation may help to maintain population differentiation. Resident populations showed evidence

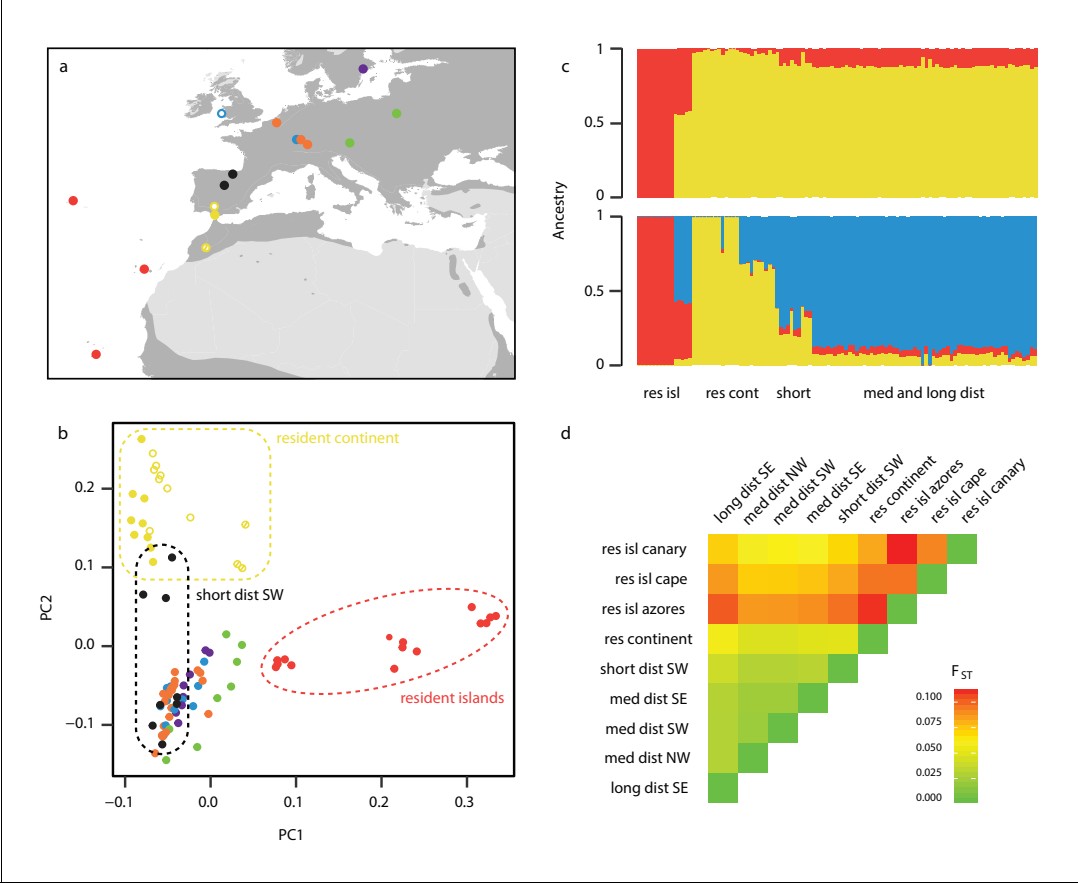

**Figure 1.** Sampling design and population structure. (**a**) Sampling sites and migratory phenotypes. Samples were collected from the breeding grounds, except for a subset of NW migrants that were sampled during winter in the UK (open blue circle) (details in *Supplementary file 4*). (**b–d**) Population structure represented by a Principle Component Analysis (PCA) (b), NGSadmix (K = 2 and 3 shown) (c) and pairwise estimates of $F_{ST}$ (d), showing differentiation between migrants and residents (as well as among residents themselves). Long dist SE = long distance migrants that orient SE in autumn (purple), med dist = medium distance migrants that orient in the corresponding heading during autumn migration (SE = green, SW = orange and NW = blue), res continent = residents found on the continent (yellow), short dist SW = short distance migrants that orient SW (black), res isl = resident birds on islands (cape = Cape Verde, canary = Canary Islands). Among continental residents, open circles indicate Cazalla de la Sierra, open circles with dash Asni, and filled circles Gibraltar. A PCA excluding islands can be found in *Figure 1—figure supplement 1*; results from NGSadmix at larger values of K can be found in *Figure 1—figure supplement 2*.

The online version of this article includes the following figure supplement(s) for figure 1:

**Figure supplement 1.** Principal component analysis matching that in *Figure 1* but excluding island populations.

**Figure supplement 2.** Complimentary figure to *Figure 1c*, showing ancestry proportions estimated by ADMIXTURE at larger cluster values (k = 4 through 7).

of historical bottlenecks followed by sudden expansions, suggesting that blackcaps lost their ability to migrate after secondary colonization of mild areas in southern Europe and on the Atlantic islands. This finding was supported by *Voelker and Light (2011)* who used mitochondrial (ND2 and cytb) data to reconstruct ancestral states within the genus *Sylvia*. Limited genetic differentiation between blackcaps was also documented by *Dietzen et al. (2008)* using mitochondrial (cytb) data; these authors also estimated dates for the colonization of Atlantic islands and for an earlier colonisation of the Canary Islands (the latter occurring 300,000–3,000,000 ya vs. 4,000–40,000 ya for colonisation of other islands including the Azores and Cape Verde).

The phylogeographic studies described above provided important insight into the evolution of migration in blackcaps and other temperate species more generally. When these studies and experimental work are considered together, the thorough set of studies conducted on blackcap migration is arguably unequalled in other species. Surprisingly, these classic experiments and molecular marker-based approaches have been followed by a dearth of genetic work on migration in

blackcaps. Here, we leverage our knowledge of this excellent study system by using high-throughput sequencing techniques to provide the first genome-wide characterization of the blackcap. The major objectives of this study were to assemble a high-quality reference genome de novo, and to use whole genome resequencing data from 110 blackcaps (including birds from each migratory phenotype and encompassing the entire breeding range, *Figure 1a*) to (1) examine population structure and demography in this system, and (2) study the genetic basis of three migratory traits in unison: the propensity to migrate, migratory distance and orientation.

Analyses of population structure and demography revealed novel insights that are important for understanding both the evolutionary history of migration in blackcaps and the underlying genetics of this behaviour. A small number of studies on the genomics of migration have been conducted in songbirds (*Delmore et al., 2016*; *Lundberg et al., 2017*). We compare our results to theirs, evaluating a long held hypothesis of a common genetic basis to migratory behaviour (*Liedvogel et al., 2011*). Our results are not only relevant to understanding the genetics of migration in the blackcap. Data on the genetics of complex behaviours is at a premium in the evolutionary literature, which has focused primarily on morphological traits, and migration probably plays an important role in the early stages of speciation in many systems. Our results will speak to the genetic basis of this process.

## Results and discussion

### Assembly of a high-quality draft reference genome

We used whole genome sequencing (WGS) data and an optical map (Illumina and Bionano Irys technology, *Supplementary files 1–3*) to de novo assemble a hybrid reference genome for the blackcap (BioProject number PRJNA545868; Guojie Zhang, personal communication). The final genome is 1.02 Gb in length, comprised of only 96 scaffolds and has a large $N_{50}$ scaffold length of 22 Mb. Ninety scaffolds mapped to the collared flycatcher *Ficedula albicollis* genome (average three scaffolds/chromosome; *Supplementary file 3*) and our annotation strategy, which used both in silico and evidence-based approaches, identified 17,982 protein-coding genes. Results from BUSCO and an analysis of UCEs (ultra-conserved elements) suggest that our reference is nearly complete, with 92% of single-copy orthologues unique to birds and 97% of UCEs identified in amniotes (*Faircloth et al., 2012*; *Supplementary file 2*).

### Population structure and demography

We aligned WGS data from 110 blackcaps (including the two birds used in our assembly) to this reference (average coverage 17.5x, *Figure 1a*, *Supplementary file 4*) and estimated genotype likelihoods at genome-wide single nucleotide polymorphisms (SNPs). Genomic differentiation was low between migratory populations of different distances and orientations, but unlike earlier work using mitochondrial data (*Pérez-Tris et al., 2004*; *Dietzen et al., 2008*), we documented considerable differentiation between migrant and resident populations on both the continent and islands. A PCA separated resident island birds from continental populations on PC1, and resident continental birds from migrants on PC2 (*Figure 1b*). Migrants were not clearly distinguished on either PC (we obtained the same result when we re-ran the PCA excluding islands; *Figure 1—figure supplement 1*). Results from an ADMIXTURE analysis and estimates of $F_{ST}$ confirm this pattern. At a cluster value of two, ADMIXTURE distinguished between island and continental birds (similar to PC1). At a cluster value of three, populations on the continent were further divided into resident and migratory groups (similar to PC2), and resident island and continent birds showed some admixture with the migratory group (*Figure 1c*). No further structure was observed beyond these three clusters (*Figure 1—figure supplement 2*). Estimates of $F_{ST}$ ranged from 0.018 to 0.11, with the highest estimates occurring between migrants and both resident groups (0.06–0.11 for islands, 0.042–0.05 for continent residents; *Figure 1d*). Evidence for limited population differentiation combined with dramatic differences in the migratory behaviour of blackcaps is ideal for identifying genomic regions that are associated with this focal trait. Specifically, genomic regions associated with migration should standout against this backdrop of limited differentiation, although analyses involving residents will need to account for elevated values of differentiation.

A phylogeographic analysis using mitochondrial data suggested that variation in migratory behaviour evolved recently, 4000–13,000 ya (*Pérez-Tris et al., 2004*). Our results move this date further back in time, to 30,000 ya and the start of the last glacial maximum (*Clark et al., 2009*). Specifically, we used multiple sequentially Markovian coalescent (MSMC, implemented in MSMC2) (*Schiffels and Durbin, 2014*; *Malaspinas et al., 2016*) to characterize the demographic history of blackcaps. The demographic trajectories of migratory, resident continent and resident island birds began to diverge ~30,000 ya. The effective population size of migrant and resident island populations expanded and contracted, respectively, while continental residents exhibited a relatively constant effective population size (*Figure 2a*; *Figure 2—figure supplement 1*). Relative cross-coalescence rates (CCR) between all three groups exhibited a concomitant drop ~30,000 ya (*Figure 2b*; *Figure 2—figure supplement 2*). The drop of relative CCR between migratory and resident island populations was steeper than that between migratory and resident continent populations (*Figure 2b*), suggesting that genetic separation following the colonization of islands resulted in greater separation than that between continental populations of migrants and residents. Increased differentiation between migrants and resident island birds (vs. resident continent birds) was also documented in our PCA (*Figure 1*). Results for medium-distance migrants (NW, SW and SE) and long-distance migrants are indistinguishable (*Figure 2—figure supplement 4*; *Figure 2—figure supplement 5*; *Figure 2—figure supplement 6*).

One interesting finding from our demographic analyses is that of apparent gene flow between migrant and resident continent birds ~5000 years ago. Specifically, the relative CCR between migrant and resident continent populations started to increase at ~5000 ya (~25,000 years after initial divergence; *Figure 2—figure supplement 3*). This admixture event may reflect secondary contact between migrant and resident continent populations and is line with our results from ADMIXUTRE, with admixture documented between these groups at a cluster value of three. The last glacial maximum ended ~19,000–11,500 ya (*Clark et al., 2009*). After this time, populations would have expanded out of their glacial refugia, and perhaps migrant and resident continent populations came into secondary contact ~5000 years after these expansions began. Similar to our results on population differentiation, island populations exhibit their own evolutionary trajectories following divergence. This result is in line with results from *Dietzen et al. (2008)*, who suggested that at least one separate colonization to the Atlantic islands occurred (earlier than that to the Canaries).

## Genetic basis of migratory traits

Here, we transition to study local patterns of genomic differentiation, identifying specific genomic regions that have signatures of selection related to three phenotypes: the propensity to migrate, orientation of migration and distance of migration (resident continent, short distance SW, medium distance NW, SW, SE and long distance SE populations). We excluded resident island birds from these analyses (because of limited sample size [n = 5] and potential effects from founder events) and focused on a single resident continent population (Gibraltar, we obtained similar results using Cazalla de la Sierra, total number of birds included in these analyses = 82, *Supplementary file 4*).

Positive selection was more common in residents and limited to a few, small genomic regions (*Table 1a*). For example, hapFLK is a tree-based method that controls for hierarchical population structure. Global and local NJ trees are constructed using haplotype frequencies and regions under selection show longer branch lengths. Only nine regions were found to be under selection (permutation test, see 'Materials and methods') according to this method, and six of these appeared in residents. *Figure 3a* shows estimates of hapFLK for the entire genome, and *Figure 4a* exemplifies one region on Super-Scaffold 99 (syntenic with flycatcher chromosome 3). The average size of these regions was 16.7 kb and only six genes occurred within them. We used CAVIAR (*Rochus et al., 2018*) to identify variants showing strong associations with selection in these regions. Each region included one to four variants, all of which occurred in non-coding regions (*Supplementary file 5*). Previous phylogeographic work suggested that migration is the ancestral state in blackcaps (*Pérez-Tris et al., 2004*; *Voelker and Light, 2011*). Accordingly, the selection in genomic regions that we identified here is probably involved in the transition from migrant to resident phenotypes.

We complemented results from hapFLK with a modified version of the Population Branch Statistics (PBS) (*Yi et al., 2010*) and the number of segregating sites by length (nSL) (*Ferrer-Admetlla et al., 2014*). PBS is an $F_{ST}$-based statistic that estimates allele frequency differences between three or more populations. This parameter can be elevated by linked purifying selection (or

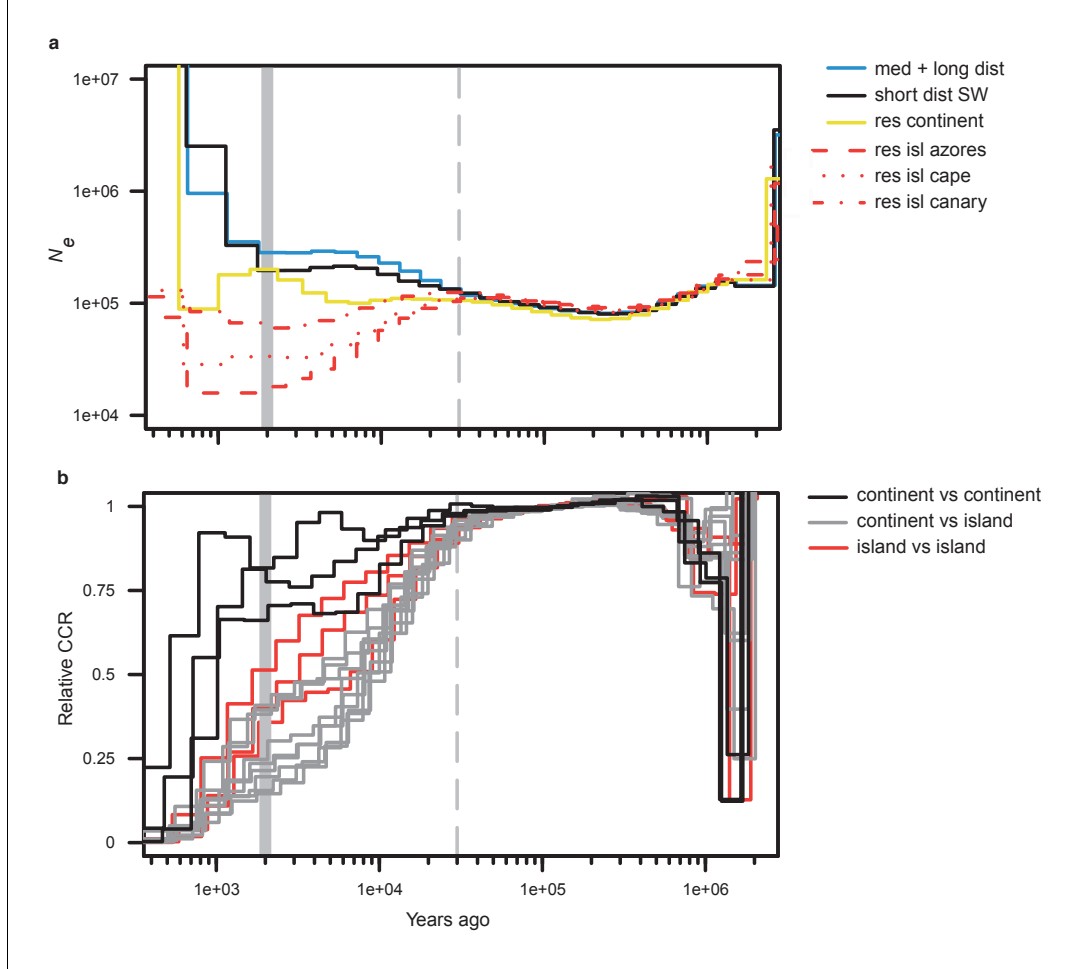

**Figure 2.** Demographic history. (**a**) Effective population size by time estimated by MSMC2 using five individuals per blackcap phenotype. Note that the most recent time segment is regarded as being unreliable in MSMC2 results. (**b**) Relative cross-coalescence rate estimated by MSMC2. 15 lines with three colours indicate relative cross-coalescence rate for all pairwise combinations of the six populations (three for comparisons between populations on the continent [continent vs. continent], three for comparisons between populations on the islands [island vs. island], and nine for comparisons between continent and island populations [continent vs. island]). The dotted vertical line indicates the inferred time of population separation. Results from down-sampling can be found in *Figure 2—figure supplements 1*, *2* and *3*; results for medium- and long-distance migrants run separately can be found in *Figure 2—figure supplements 4*, *5* and *6*.

The online version of this article includes the following figure supplement(s) for figure 2:

**Figure supplement 1.** Down-sampling for demography analysis of effective population size.

**Figure supplement 2.** Down-sampling for demography analysis of relative cross-coalescence rate.

**Figure supplement 3.** Demography analysis of relative cross-coalescence rate.

**Figure supplement 4.** Medium distance NW, SW and SE migrants and long distance migrants show similar demographic histories.

**Figure supplement 5.** Medium distance NW, SW and SE migrants show similar demographic histories.

**Figure supplement 6.** Medium distance NW, SW and SE migrants show similar demographic histories.

background selection) within populations that is unrelated to positive selection (in our case selection related to migration). We removed these confounding effects by scaling PBS and subtracting the maximum value of PBS in orthologous windows from that in the non-focal population (hereafter 'ΔPBS', following *Vijay et al., 2017*). nSL is a haplotype-based statistic that focuses on patterns within populations, using segregating sites to measure the length of haplotypes. Linked selection should increase haplotype lengths at genomic regions that are under positive selection. Eight of the nine regions identified by hapFLK also exhibited extreme values of ΔPBS and nSL (in the top 1% of the distribution) in the same population as that identified by hapFLK (ΔPBS *Table 1a*, *Figure 3b* for resident birds [estimates for short distance SW, medium distance SE, SW, NW, and long distance

**Table 1.** Genetic variants underlying variation in migration.

(a) Results from analyses including all continental birds and (b) results from analyses limited to medium-distance migrants. Results from hapFLK include the size, the population where the signal was found and genes within the region. Estimates of ΔPBS and (PBS) in the same regions are shown; they are bolded if in the top 1% of the focal population's distribution and new sizes are estimated using neighbouring windows above this threshold (if larger than the limits from hapFLK, additional genes are specified). Estimates of PBS were re-estimated using island populations (vs.continent resident populations). Regions in the top 1% of an island population's distribution are indicated in section (a) (recorded as 'NA' if the initial population under selection was not resident). 'Scaf' refers to the scaffold within the blackcap genome where the region is found and 'chr' refers to the flycatcher chromosome that these scaffolds map to. For the number of strongly associated SNPs identified by CAVIAR and estimates of nSL, see *Supplementary file 5*.

(a)

| | | hapFLK | | | | Δpbs | | | |
|---|---|---|---|---|---|---|---|---|---|
| Scaf | Chr | Size (bp) | Log p-value | Population | Genes | Size (Mb) | ΔPBS (PBS) | Island replacement | Genes |
| 12 | 4A | 14,059 | 9.4 | Resident | LOC100859173 | 52 | **18.7 (0.40)** | Azores | EDA2R |
| 13 | 11 | 29,195 | 8.3 | Resident | CHST4, TERF2IP, KARS | 303 | **41.0 (0.87)** | Cape Verde | DHX38, DHODH, IST1, C2H2, ATXN1, AP1G1, PHLPP2, TAT, GABARAPL2, TMEM231, CHST6 |
| 17 | 3 | 7610 | 9.5 | Short SW | | 316.5 | **0 (0.02)** | NA | NKAIN1 |
| 22 | 9 | 53,890 | 8.8 | Med SE | CLSTN2 | 1,005.5 | **21.9 (0.19)** | NA | DUF4637, PIK3CB, FOXL2, MRPS22, COPB2, RBP2, NMNAT3 |
| 30 | 2 | 13,756 | 11.5 | Resident | | 42.5 | **8.14 (0.19)** | Cape Verde | |
| 30 | 2 | 7902 | 8.8 | Resident | | 1,029.5 | **19.1 (0.42)** | Canaries, Cape Verde | |
| 41 | 8 | 10,341 | 8.3 | Resident | | 11.5 | **15.0 (0.33)** | | |
| 46 | 1A | 412 | 7.9 | Med SE | | 9.5 | **9.0 (0.03)** | NA | |
| 99 | 3 | 13,140 | 7.8 | Resident | TTBK1 | 192 | **28.6 (0.61)** | Azores, Canaries, Cape Verde | LOC101820716, ACSS1, NEIL1, SLC22A7, TTL |

(b)

| | | hapFLK | | | | Δpbs | |
|---|---|---|---|---|---|---|---|
| Scaf | Chr | Size (bp) | Log p-value | Population | Genes | Size (Mb) | ΔPBS (PBS) |
| 17 | 3 | 3258 | 9.04 | Med NW | SDC1 | 5 | 14.49 (0.20) |
| 30 | 2 | 311 | 8.85 | Med NW | | 7 | 11.31 (0.16) |
| 46 | 1A | 461 | 8.71 | Med NW | | 3 | 8.14 (0.15) |
| 63 | 1A | 1088 | 9.55 | Med SE | | | 1.14 (0.05) |
| 67 | 6 | 995 | 9.46 | Med SW | | 5 | 9.03 (0.18) |
| 73 | 5 | 3611 | 11.81 | Med NW | ATG2B, BDKRB2 | 3 | 30.41 (0.35) |

SE, *Figure 2—figure supplement 4*; *Figure 2—figure supplement 5*; *Figure 2—figure supplement 6*]; nSL *Supplementary file 5*).

As noted already, population structure and linked selection can elevate differentiation between populations. We controlled for these effects using hapFLK and ΔPBS, respectively, and emphasise that genomic differentiation between populations of blackcaps is low to begin with (*Figure 1d*). In addition, linked purifying selection would be expected to increase PBS in all populations (i.e., not just the focal population), but this is not the case. This is exemplified in *Figure 5a* where estimates of ΔPBS for all populations are shown but these are only elevated in the resident continent population. As a final test of population structure, we re-estimated ΔPBS using resident island birds (instead of resident continent birds). We conservatively excluded these populations from our initial analyses because their sample sizes are small and because genetic drift can affect estimates of differentiation in island populations. Nevertheless, the island populations are also resident and thus these estimates could help to validate the genomic regions that were identified as being under selection in resident populations on the continent. *Table 1a* summarizes these results, noting which genomic regions

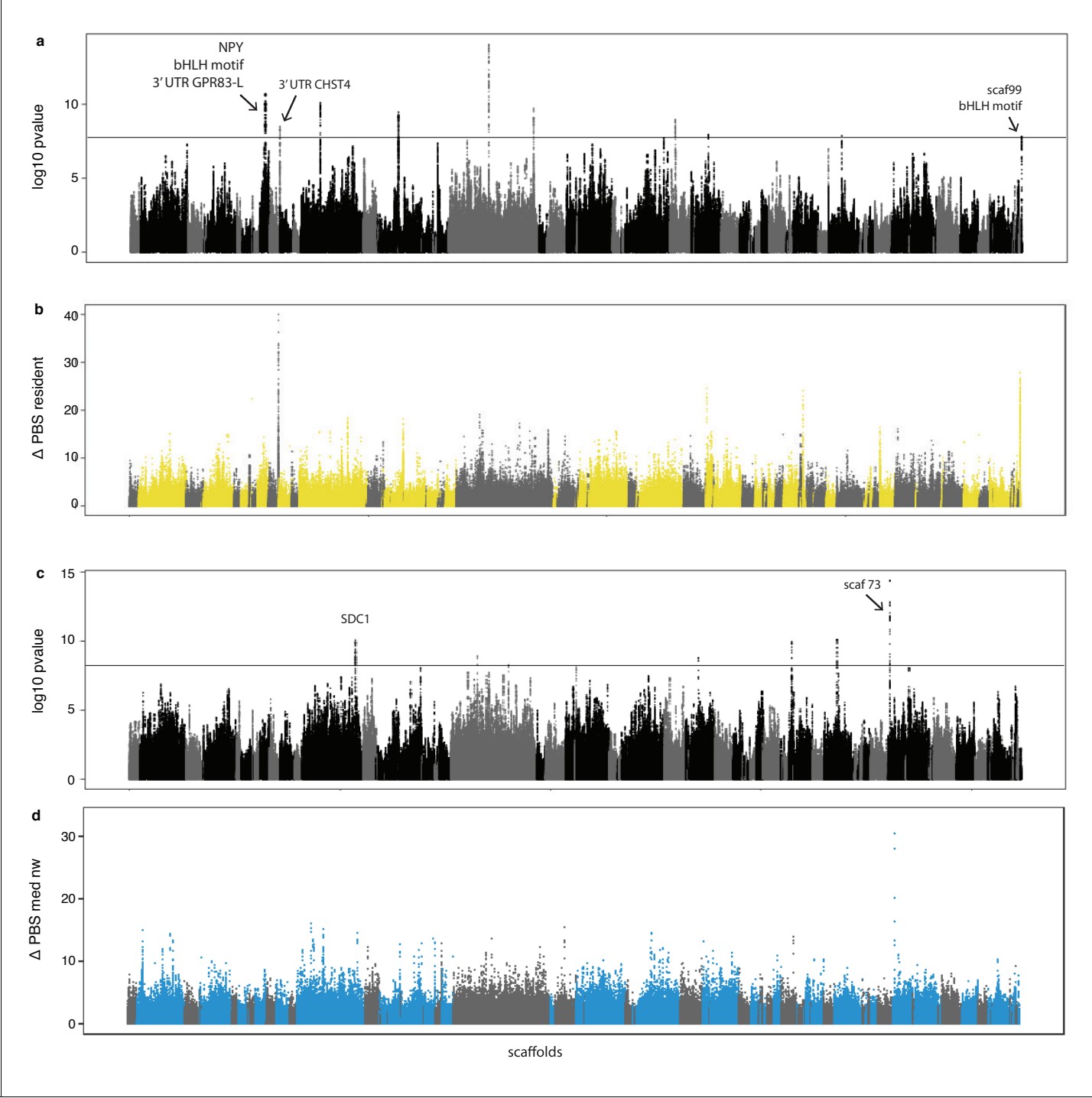

**Figure 3.** Genome-wide local estimates of population differentiation. Results from hapFLK using haplotype frequencies (**a,c**) and ΔPBS using SNP frequencies (**b,d**; 2,500 bp windows). Estimates of ΔPBS for resident continent (**b**) and medium-distance NW migrants (**d**) are shown; results for the remaining populations can be found in *Figure 3—figure supplement 1*. Genetic elements, scaffolds and genes discussed in the text are highlighted. The online version of this article includes the following figure supplement(s) for figure 3:

**Figure supplement 1.** Genome-wide local estimates PBS for the remaining populations.

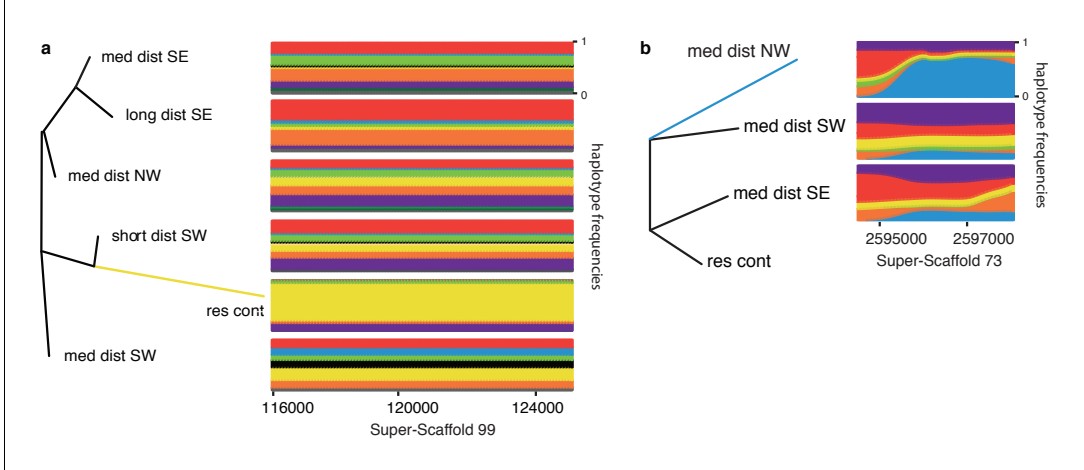

**Figure 4.** Exemplifying genomic regions under positive selection. Local neighbour joining trees for regions under selection in (**a**) the resident continent population on Super-Scaffold 99, and (**b**) medium-distance NW population on Super-Scaffold 73. Selection is indicated by longer branch lengths in each population than is the case in global trees built using data from all genomic regions (*Figure 4—figure supplement 1*). Panels to the right of the trees show the corresponding frequency of haplotypes in each population of the tree. Haplotype clusters are colour coded (colours of haplotype clusters do not correspond to the population colour coding used in other figures), and frequencies are plotted along the Y axis. Haplotype frequency plots show the near fixation of a single dominating haplotype in (**a**) resident continent (yellow) and (**b**) medium-distance NW populations (blue). The location (in bp) of these regions on each Super-Scaffold is shown below these panels and the resident continent group is only included to root the tree in panel (**b**), and thus has no haplotype frequencies.

The online version of this article includes the following figure supplement(s) for figure 4:

**Figure supplement 1.** Global neighbor joining trees built using hapFLK data and data from all genomic regions, for comparison with local trees showing positive selection in *Figure 4*.

exhibited elevated values of PBS on islands. Of particular interest, PBS was elevated in all three island populations at the genomic region on Super-Scaffold 99 (*Figure 5b*). Combined with findings from hapFLK (controlling for population structure and relying on haplotypes), ΔPBS (controlling for linked selection and relying on SNP data) and nLS (estimated within populations and relying on haplotypes), these results provide strong evidence that this specific region contains important variation for the transition to residency, not only on the continent but also on the islands.

Note that it is possible that the signatures of positive selection that we document here reflect selection based on different ecological variables involved with the colonization of areas further south on the continent, but at least in the case of Super-Scaffold 99, we believe that this is rather unlikely as most ecological variables (biotic and abiotic) are quite distinct between islands and the continent (and between the islands themselves) (*Cropper, 2013*; *Valente et al., 2017*). The transition to residency is shared, probably representing one of the only shared selection pressures experienced by all of these populations. Note that the lack of consistent results for other regions under selection in the resident continent population does not preclude the potential importance of these regions as, for example, genetic drift on islands would affect which genetic variants were present on islands for selection to act on.

Our finding that only a few genomic regions under selection contain genes and that the strongly associated SNPs identified by CAVIAR are in non-coding regions could suggest that cis-regulatory changes are important for the transition from migration to residency. In support of this suggestion, an alignment of predicted mRNAs from several bird species and transcripts from a testis transcriptome of the blackcap placed two of the SNPs from CAVIAR in the 3′ untranslated region (3′ UTRs) of two genes (GPR83-L on Super-Scaffold 12 and CHST4 on Super-Scaffold 13, syntenic with flycatcher chromosomes 11 and 4a, respectively). Three prime3′ UTRs can act as posttranscriptional regulators; they contain binding sides for microRNAs, which can inhibit translation or target mRNA for degradation (*Mayr, 2017*; *Barrett et al., 2012*). In fact, previous work with monarch butterflies identified 55 conserved microRNAs that are differentially expressed between summer and migratory butterflies (*Zhan et al., 2011*). Future analyses to validate this suggestion could include the use of qPCR to

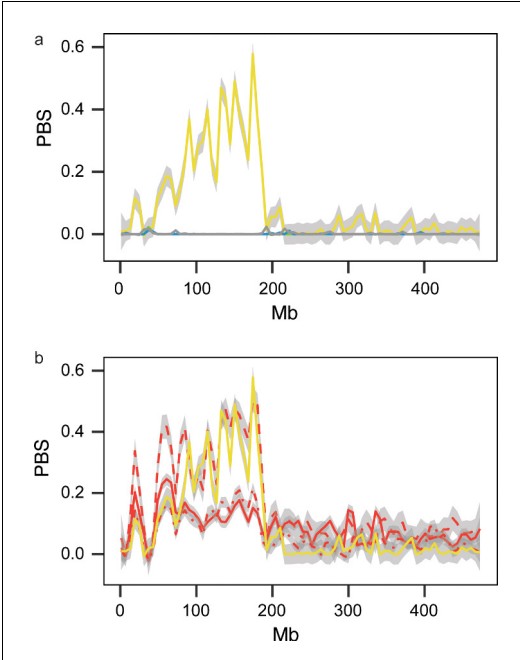

**Figure 5.** Estimates of ΔPBS on Super-Scaffold 99 corresponding with the region shown in *Figure 4a* (smoothed using the geom_smooth function in ggplot to summarize data in 2500-bp windows). (a) Estimates for resident continent, medium-distance NW, SW and SE migrants, and short- and long-distance birds. These estimates are only elevated in the resident continent phenotype, ruling out a role for linked selection in generating this signature in residents. (b) Estimates for the resident continent and island birds (Azores, Canaries and Cape Verde), which are all elevated, implying that parallel selection is probably involved in the transition from migration to residency in this region. Colours correspond to *Figure 1a* with yellow showing data for resident continent birds.

determine whether GPR83-L and/or CHST4 are in fact differentially regulated between the migratory phenotypes.

Future work using techniques aimed at identifying binding sites for transcription factors (e.g., ChIP-seq) could also be useful. We conducted a preliminary analysis here, using HOMER (*Heinz et al., 2010*) to detect known transcription factor motifs in the genomic regions that are under selection in residents. Specifically, *Ruegg et al. (2014)* used a literature search to identify 25 candidate genes for migration. Four of these genes are transcription factors whose motifs are in the libraries searched by HOMER: three basic helix-loop-helix transcription factors (bHLH) (*Clock*, *Npas2*, and *Bmal1*) and one basic leucine zipper domain (*Nfil3*). We found a bHLH motif (GHCACGTG) on Super-Scaffolds 12 and 99 (*Figures 3a,b*, *4a* and *5*). The motif on Super-Scaffold 99 is particularly interesting as there is a SNP (G/T) at the beginning of the motif that is nearly fixed in continental residents (the allele frequency for G in Asni, Gibraltar and Cazalla de la Sierra is 1, 0.85 and 0.9, respectively; $F_{ST}$ between Gibraltar and medium-distance NW, SW and SE migratory populations is 0.15, 0.25 and 0.44, respectively). This motif could disrupt or weaken transcription factor binding (*Kasowski et al., 2010*). This is also the genomic region that showed elevated PBS in both resident continent and island populations (*Figure 4a*, *Figure 5*). *Clock*, *Npas2* and *Bmal1* are involved in maintaining circadian rhythms. Circadian rhythms synchronize circannual clocks, which are important cues controlling seasonal migratory behaviour (*Gwinner, 1996*; *Visser et al., 2010*).

Concerning the actual identity of genes within regions that are under selection, several have functions that could be related to the transition from migration to residency. For example, LOC100859173 (located on in the genomic region under selection on Super-Scaffold 12, the region with a bHLH motif mentioned above; *Table 1a*) has been annotated as a probable G-protein coupled receptor that mediates the function of neuropeptide Y (NPY). NPY is localized in the brain of birds and works with Agouti-related peptide (AGRP) and proopiomelanocortin (POMC) to control energy balance. Specifically, NPY/AGRP neurons stimulate appetite, food intake and fat deposition, while POMC inhibits these processes (*Boswell and Dunn, 2017*). It has been hypothesized that the effects of NPY may extend to seasonal changes in energy balance that are important for migration, including hyperphagia and fat deposition (*Boswell and Dunn, 2017*). Beyond its role in energy balance, NPY also facilitates learning and memory via the modulation of hippocampal activity and has an effect on circadian rhythms, reproduction, and the contraction of vascular smooth muscles. It has been suggested that a common genetic mechanism or major regulator may control migratory traits (*Liedvogel et al., 2011*; *Liedvogel and Lundberg, 2014*). A protein such as NPY, or the transcription factors that bind the bHLH motif identified in the prior analysis, could fill this role.

## Analysis focused on migratory orientation and distance

So far, we have considered all three migratory traits exhibited by blackcaps together (propensity, orientation and distance) and our results relate mostly to residents. The elevated population differentiation that we noted between resident and migratory birds could reduce our power to identify selection that is specific to migrants (*Fariello et al., 2013*). Accordingly, we ran a second set of analyses excluding resident birds and examining migratory orientation and distance independently. Starting with orientation and limiting our analysis to medium-distance migrants with varying orientations (medium-distance NW, SW and SE migrants, total number of birds included in these analyses = 54, *Supplementary file 4*), hapFLK identified only six regions that are under positive selection (*Table 1b*). Most of these regions showed selection in the NW phenotype and exhibited extreme values of ΔPBS limited to the population identified by hapFLK (*Figure 3c,d*). *Figure 4b* exemplifies results for hapFLK at one region under selection in the NW migrants (~4 kb on Super-Scaffold 73, syntenic with flycatcher chromosome 5). Results for nSL can be found in *Supplementary file 5*.

The list of genes in genomic regions that are under selection in this analysis focusing on orientation is small, but it also includes genes with functions that are strongly related to the phenotype they are associated with. For example, SDC1 is a region on Super-Scaffold_17 that is under selection in NW migrants. This gene codes for a transmembrane protein that helps to regulate the Wnt signalling pathway. This pathway plays a role in embryonic development and has been shown to influence feather and beak morphogenesis, along with feather molt (*Yu et al., 2004*; *Mallarino et al., 2011*; *Bhullar et al., 2015*; *Widelitz, 2008*). NW migrants have rounder wings and more narrow beaks than southern migrants (*Rolshausen et al., 2009*). Differences in the timing of migration probably mean that birds also molt at different times. This has not been evaluated directly in comparisons between migrants, but variation in molt patterns have been documented between NW migrants and birds that are resident on the continent (*de la Hera et al., 2009*).

Two previous studies attempted to identify de novo genomic regions under selection related to differences in orientation: *Delmore and Liedvogel (2016)* with Swainson's thrushes (*Catharus ustulatus*) and *Lundberg et al. (2017)* with willow warblers (*Phylloscopus trochilus*). *Delmore and Liedvogel (2016)* identified a region on chromosome 4 and *Lundberg et al. (2017)* regions on chromosome 1 and 5 that are associated with orientation. None of these regions overlap with those under selection in our study on blackcaps. It is tempting to suggest that migration may be controlled by similar genes across broad taxonomic scales, with early results from candidate genes (e.g., the poly-glutamine repeat in *Clock*) showing consistent results across groups as divergent as insects, fishes and birds (*Delmore and Liedvogel, 2016*). Nevertheless, several studies have failed to document an association with *Clock,* and a comparison of our results with those of *Delmore and Liedvogel (2016)* and *Lundberg et al. (2017)* adds further caution to this idea of a common basis (at least at the sequence level). This is an important finding as it has long been hypothesized that there may be a shared genetic mechanism for migration, not only in birds but also in other taxonomic groups (*Liedvogel et al., 2011*; *Liedvogel and Lundberg, 2014*; *Liedvogel and Delmore, 2018*).

None of the regions identified by hapFLK and PBS were fixed for alternate haplotypes or alleles. This fact is evident in *Figure 4*, in which the regions under selection still include haplotypes from a different cluster, and it could suggest that selection is acting on shared genetic variation (i.e., variation that is already present in the population rather than newly derived mutations). The idea that transitions between migratory phenotypes have been facilitated by shared genetic variation has been around for quite some time in the blackcap literature, particularly as rapid transitions have been observed and include the evolution of a new NW migratory route in the past 70 years. Shared variation can facilitate these rapid changes as these variants are already present in the population and have been tested by selection (*Barrett and Schluter, 2008*). The fact that regions under selection are quite narrow (*Table 1*) also supports a role for shared genetic variation (*Barrett and Schluter, 2008*) and we provide further evidence below.

First, we estimated the genetic distance between one haplotype in each cluster and an ancestral sequence that we derived using WGS from the two most closely related sister taxa, hill babbler (*Pseudoalcippe abyssinica*, an African resident) and garden warbler (*Sylvia borin*, a long distance migrant) (*Voelker and Light, 2011*). Using the region on Super-Scaffold 73 that shows selection in NW migrants (*Figure 4b*), we predicted that if haplotypes in the light blue cluster were present in the population already, they should exhibit similar levels of divergence from the ancestral sequence

as haplotypes from all other clusters. This is precisely what we found; genetic distance from the ancestral sequence was similar for haplotypes from all clusters (181 differences for the NW haplotype vs. 178, 179 and 181 [x3] and 182 differences in the rest). We reran this analysis limiting our data to synonymous substitutions in predicted coding regions (i.e., those that are likely to be evolving neutrally and located in ATG2B and BDKDB) and obtained similar results. Specifically, we identified six synonymous substitutions between all three medium-distance migrant populations and both garden warblers and hill babblers, suggesting that there is no difference in the age of these haplotypes.

To follow up on the former analysis, we constructed a maximum likelihood (ML) tree using sequence data from the region under selection on Super-Scaffold 73. We built this tree using data from all continental blackcaps, garden warblers, and hill babblers, using the willow warbler as an outgroup, and compared this tree to a consensus tree summarizing ML trees constructed for each scaffold in the blackcap reference genome (i.e., a tree built using genome-wide data; *Figure 6a*). Supporting previous phylogenetic work in the system, garden warblers and hill babblers formed a sister clade to blackcaps in the consensus tree, and relationships among blackcaps were largely unresolved. By contrast, garden warblers were more closely related to blackcaps than were hill babblers in the tree built using data from the region on Super-Scaffold 73 (*Figure 6b*). In addition, the medium-distance NW population (in which positive selection is acting in this particular region) occurs at the base of the blackcap clade. Recall that garden warblers are obligate migrants whereas hill babblers are residents, supporting the suggestion that haplotypes favoured in the NW phenotype were already present in the population before divergence, perhaps even in ancestral populations. Unfortunately we do not have data from any closely related species to determine how old this haplotype is (i.e., if it is older than the split between garden warblers, hill babblers and blackcaps sensu *Colosimo et al., 2005*).

In a final analysis, we compared the site frequency spectrum (SFS) for the region on Super-Scaffold 73 to SFSs estimated for 1000 random sequences of the same length from throughout the genome. SFSs for the random sequences are similar to expectations under neutrality, with a preponderance of alleles at low frequencies. By contrast, the SFS of Super-Scaffold 73 shows an excess of mid-frequency alleles (*Figure 6c*). Greater variance in SFSs are expected when selection makes use of standing variation because alleles have been recombining onto different backgrounds in ancestral populations (*Przeworski et al., 2005*; *Pennings and Hermisson, 2006*).

We conclude our study by examining the genetic architecture of migratory distance. We included all migrants in this analysis, quantified migratory distance as an ordinal variable from short- (1), to medium- (2), to long-distance (3) migrants, and used a Bayesian Sparse Linear Mixed Model (BSLMM, 87) to identify SNPs that are associated with migratory distance (total number of birds included in these analyses = 72, *Supplementary file 4*). BSLMMs are a form of genome-wide association analysis that includes a term for other factors that influence the phenotype and are correlated with genotype (e.g., population structure and ancestry; a kinship matrix based on genome-wide SNP data) and can be used to estimate both the combined effects of multiple SNPs and the effects of SNPs on their own.

Our results suggest that a large percentage of variance in migratory distance can be explained by our SNP set (PVE = 0.90 ± 0.28), but only one SNP showed a strong association with this focal trait (posterior inclusion probability >0.01). This SNP is located on Super-Scaffold 79, occurs in an area of elevated $F_{ST}$ between long- and short-distance migrants ($F_{ST}$ = 0.31, in 0.018 percentile $F_{ST}$ values) and is 627-bp downstream from the gene KCNIP1, which encodes a potassium channel interacting protein (major determinants of neuronal cell excitability). Combined with the haplotype identified in the hapFLK analysis, which provides a signature of positive selection in short-distance migrants on Super-Scaffold 17 (*Table 1a*), these loci represent good candidates for controlling migratory distance, but future analyses with a larger sample size are needed to confirm the robustness of this finding. Direct information on migratory distance could also inform this analysis by allowing us to code the phenotype as continuous.

## Conclusions

Early research on blackcaps was pivotal for demonstrating the existence of a genetic basis of migration and studying its evolution. This is due in large part to the tractability of this species and its variability in migratory behaviour. Here, we have expanded this study system beyond phenotypic and marker-based approaches, launching it into the genomic era and conducting one of the most

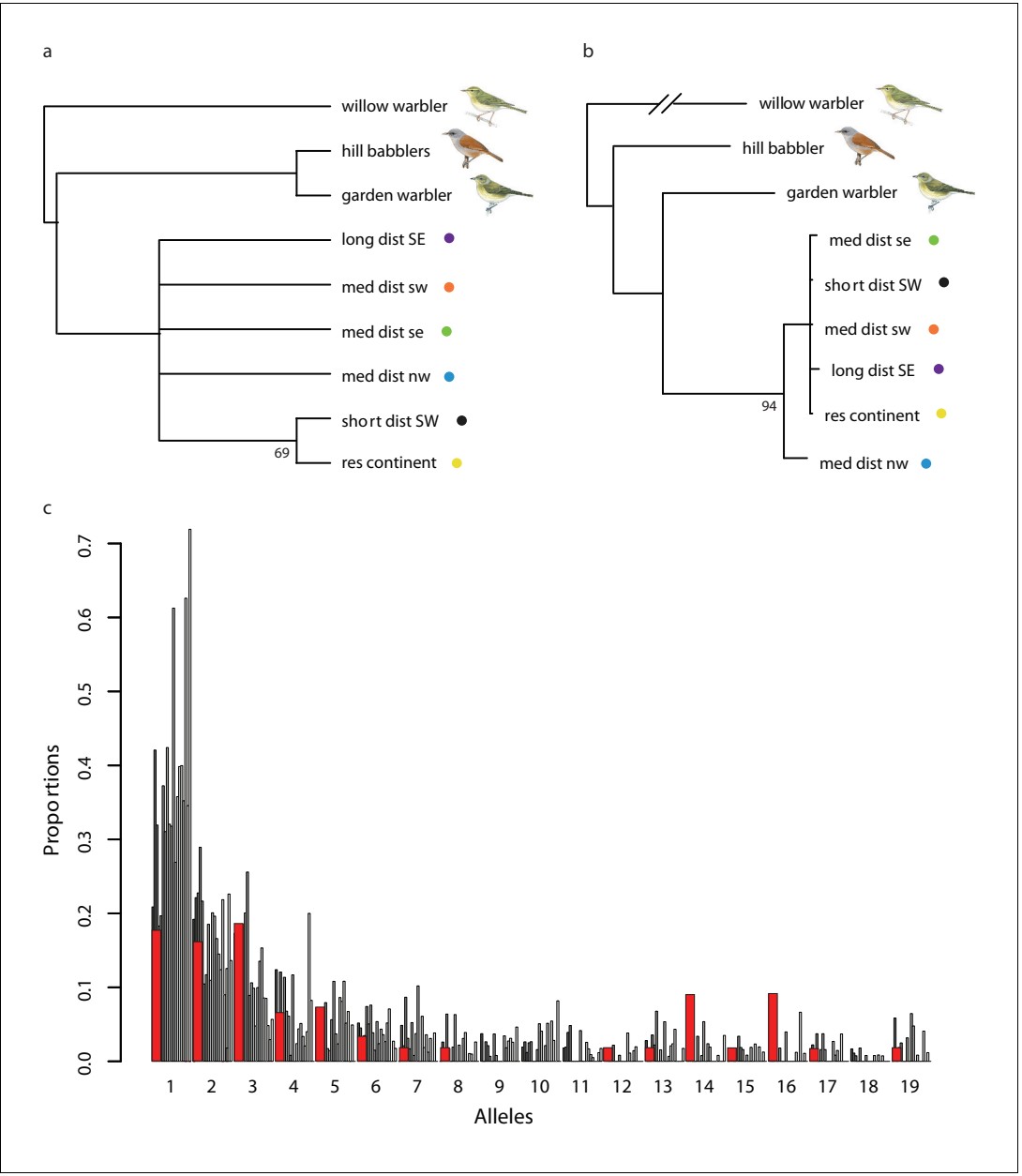

**Figure 6.** Evidence for the use of shared variation on Super-Scaffold 73. (**a**) A rooted extended majority rule consensus tree summarizing maximum likelihood (ML) trees constructed for all scaffolds in the blackcap reference genome (96 scaffolds). Node numbers indicate the number of scaffolds in which populations were partitioned into two sets. (**b**) A ML tree constructed for the region on Super-Scaffold 73 with migratory garden warbler more closely related to blackcaps and medium-distance NW birds occurring at the base of this clade in *Figure 4b*. Nodes with bootstrap values <80 are collapsed; nodes without numbers have support values of 100. (**c**) The site frequency spectrum (SFS) for the region on Super-Scaffold 73 (red) compared to SFSs for 1000 random sequences from the genome (varying shades of gray).

comprehensive genome-wide analyses of migration to date. Populations of blackcaps began to diverge ~30,000 years ago, but differentiation remains low between migratory populations. There is evidence for past gene flow between migratory and resident populations on the European continent but comparison of the contemporary structure of these populations suggests that gene flow may be limited. This is certainly the case for resident island birds. It has been suggested that one single genetic mechanism controls migratory traits and may be shared across broad

taxonomic groups. We do not find evidence for one common genetic mechanism across species here, and no protein-coding change is shared across the three focal traits (propensity, distance and orientation) that we examined in unison. Future work on gene expression may identify major regulators that control multiple migratory traits, and both NPY and bHLH transcription factors are good candidates. Combined with the additional results that we presented here (such as the importance of standing genetic variation), this information is vital for understanding how predictable the evolution of migration and other complex behavioural traits may be.

Blackcaps have not only been relevant to work on the evolution and genetics of migration. Early work in this system suggested that differences in migration might serve as reproductive isolating barriers early in speciation. For example, hybrids were shown to exhibit intermediate orientation behaviour that was predicted to be inferior because it would bring hybrids over large ecological barriers that pure forms avoid (*Helbig, 1991b*). More recently, it was shown that NW migrants arrive on the breeding grounds earlier than SW migrants, and that these birds mate assortatively on the basis of arrival time, helping to reduce gene flow between phenotypically distinct groups (*Bearhop et al., 2005*). The role of migration in speciation has gained considerable traction in recent years (*Rolshausen et al., 2009*; *Bearhop et al., 2005*; *Irwin and Irwin, 2005*; *Rohwer and Irwin, 2011*; *Turbek et al., 2018*; *Delmore and Irwin, 2014*; *Bensch et al., 2009*) and results from our study suggest that selection at a very small number of loci may be sufficient to initiate reductions in gene flow very early in the process of population differentiation and speciation.

## Materials and methods

### Genome assembly

Blood samples from two male blackcaps from the Mooswald breeding population at Freiburg im Breisgau, Germany, classified as medium-distance SW migrants (on the basis of morphometrics and isotope signatures) were used to assemble the reference genome. Full details on all steps in our genome assembly can be found in *Supplementary file 6* (BioProject number PRJNA545868; Guojie Zhang, personal communication). Briefly, genomic DNA from one individual was used to sequence Illumina sequencing libraries (fragment and mate pair libraries with insert sizes of 2, 5 and 10 kb). 275.9 Gb of raw high throughput sequence (HTS) data were generated and assembled using ALL-PATHS-LG. This assembly was improved several ways (e.g., by removing duplicates and closing gaps). DNA from the second individual was used to generate two BioNano optical maps (one using BspQI and the other BssSI). These maps were used to super-scaffold HTS scaffolds. Statistics for the final assembly and each stage can be found in *Supplementary file 2* .

We used SatsumaSynteny (*Grabherr et al., 2010*) to determine which avian chromosome each scaffold was found on (aligning scaffolds to the flycatcher genome, *Supplementary file 3*). We validated our initial ALLPATHS assembly, the improved ALLPATHS assembly and our final assembly (including BioNano optical maps) using BUSCO (version 3.0.2, AUGUSTUS species chicken and aves_odb9 dataset) and by blasting ultra-conserved elements (UCEs) identified by *Faircloth (2016)* using whole-genome alignments for the chicken and zebra finch (*Supplementary file 2*).

### Genome annotation

We annotated genes with putative functions and protein domains using MAKER. Gene prediction was performed using a de novo testis transcriptome of blackcaps and cDNAs from three avian species (zebra finch, chicken and flycatchers) from the ensembl database. Following MAKER, we obtained the predicted protein sequences to annotate genes functionally using blastp and Interproscan. For the final annotation, we only included gene predictions that either had an Annotation edit Distance (AED) <0.5 and/or a blastp hit (with the thresholds described above) and/or a predicted protein domain.

### Resequencing analysis

We obtained whole genome resequencing (WGS) data from 110 male blackcaps (including WGS data from the two individuals used to generate the reference genomes). High molecular weight DNA was extracted from blood withdrawn from the brachial vein, following a standard salt extraction protocol. Individual samples were collected across the European breeding range including three

island populations (Canary Islands, Cape Verde, and Azores) and covering the entire range of migratory phenotypes. Population phenotype was scored on the basis of morphometry, stable isotope signature and/or ringing-recovery data from selected individuals (see *Supplementary file 4* for a description of how each population was phenotyped). Birds were sampled during the breeding season unless indicated otherwise. Specifically, exceptions are a subset of UK overwintering birds (n = 6) sampled during the winter in the British Isles, and a subset of long-distance SE migrants (n = 5) caught during autumn migration and selected on the basis of wing length (see *Supplementary file 4* for details). We also obtained WGS data for five garden warblers and three hill babblers, the closest sister taxa to blackcaps, sampled during breeding (*Voelker and Light, 2011*). We prepared small insert libraries using DNA from each individual and sequenced five samples per lane on NextSeq 500 with paired-end 150 bp reads. We trimmed reads with trimmomatic (TRAILING:3 SLIDINGWINDOW:4:10 MINLEN:30) (*Bolger et al., 2014*).

All analyses made use of data from resequencing reads that were aligned to the reference genome using bwa *mem* (*Li and Durbin, 2009*) or stampy in the case of the garden warblers (divergence time of 0.026 based on alignments of UCEs (*Faircloth, 2016*; https://github.com/faircloth-lab/phyluce/). GATK (*McKenna et al., 2010*) and picardtools (http://broadinstitute.github.io/picard) were used to identify and realign reads around indels (*RealignerTargetCreator*, *IndelRealigner*) as well as remove duplicates (*MarkDuplicates*, all default settings).

We recalibrated the resulting bam files using GATK's base quality score recalibration (BQSR). Specifically, we called SNPs for each population separately using three different programs and default settings: UnifiedGenotyper from GATK, samtools (*Li et al., 2009*) and FreeBayes (*Garrison and Marth, 2012*). BQSR requires a set of known variants. We used SNPs identified in all three programs and populations as the set of known variants for the first round of BQSR. We conducted a second round using common SNPs from the three programs that were also of high quality (QUAL >995,~10% of the common SNPs).

Most of our analyses made use of the BQSR recalibrated bams, calling genotype likelihoods (GL) with ANGSD (version 0.910–24-gf84f594, *Korneliussen et al., 2014*) and filtering reads that did not map to a unique location, did not have a mapping pair, or had mapping qualities below 20 and flags $\geq$256. When it was not possible to use GL as input, we used a vcf that had been run through GATK's variant quality score recalibration (VQSR). VQSR also requires a set of known SNPs. We used the second set of known SNPs (common and high-quality) from BQSR for this analysis and combined variants from all populations into a single vcf file for subsequent analyses. All repetitive regions were excluded from our analyses and those focused on demography did not include the Z chromosome.

## Principal components analysis (PCA) and ADMIXTURE analyses

We conducted a PCA using smartpca (EIGENSOFT version 5.0) and the vcf produced from VQSR. Default parameters were used in smartpca except for the addition of a correction for LD across SNPs (nsnpldregress = 2). We conducted an admixture analysis using GLs from ANGSD and running them through ngsADMIX (*Skotte et al., 2013*) with 8 values of K (1–9).

## F<sub>ST</sub>

We estimated $F_{ST}$ between all populations using GLs from ANGSD, starting by estimating unfolded site frequency spectrums (SFS) for each population (doSaf 1, gl 1) and using them to obtain joint frequency spectrums (2DSFS, realSFS) for each pair of populations. These 2DSFSs were used as priors for allele frequencies at each site to estimate $F_{ST}$ (realSFS fst index). In order to estimate unfolded SFS, we needed an ancestral sequence, or the ancestral state of variants segregating in blackcaps. This sequence was generated using WGS from garden warblers and hill babblers. Specifically, we used samtools to generate fasta files for each garden warbler and hill babbler (n=5 and n=3, respectively) and used rules outlined in *Poelstra et al. (2014)* to call ancestral states, with alleles that were homozygous in both outgroup species being considered ancestral and excluding remaining sites (those that were triallelic or heterozygous in the outgroup species).

## Consensus tree

We obtained consensus fasta sequences for each population using ANGSD (-doFasta 2 –doCounts 1 –minQ 20 –setMinDepth 5) and used IQTREE (*Nguyen et al., 2015*) to construct maximum

likelihood trees for each scaffold in the blackcap genome (there was no difference in the topology obtained for scaffolds mapping to the Z chromosome so they were included in the consensus, data not shown). We summarized the resulting trees using phylip 'consense' and constructing an extended majority-rule consensus tree (in which nodes that were supported by fewer than 50% of the input trees are collapsed).

## MSMC2

We used MSMC2 to infer the demographic history of blackcaps in our dataset. MSMC2 implements the multiple sequentially Markovian coalescent (MSMC) model, estimating effective population size by time and relative cross-coalescence rates between any two populations. It allows inference of the expansions and contractions of a population and of the extent and timing of population divergence (*Malaspinas et al., 2016*). Specifically, by running a hidden Markov model (HMM) along all possible pairs of haplotypes, MSMC2 estimates the free parameters for a demography model (a series of effective population sizes as a function of segmented time) and relative cross-coalescence rates between sequences using a maximum-likelihood approach.

After phasing our data using fastphase (*Scheet and Stephens, 2006*), we combined individuals into six groups (medium and long migrants ['med + long'], short-distance SW migrants ('short'), resident continent birds, and resident island birds from the Azores, Cape Verde, and Canary Islands). We grouped medium (NW, SW and SE) and long-distance migrants because they exhibited very little population structure (*Figure 1*) and indistinguishable demographic histories (*Figure 2—figure supplement 4*; *Figure 2—figure supplement 5*; *Figure 2—figure supplement 6*). We excluded any birds with less than 15x coverage. This filter left us with all island individuals (five individuals for each island), five short migrants, 19 continental residents, and 44 med + long migrants. To avoid bias associated with the use of unequal numbers of individuals from each group, we randomly down-sampled five individuals from med + long migrants and continental residents to create 10 sample groups. We used the bamCaller.py script provided in the msmc-tools package (https://github.com/stschiff/msmc-tools; *Khvorykh, 2018*) to create sample-specific callability mask files. We generated a global mappability mask file for the reference genome using GEM (*Derrien et al., 2012*). We inferred effective population size by running MSMC2 separately for each group (*Schiffels and Wang, 2020*). We determined the number of clusters for fastPHASE using a cross-validation procedure (https://github.com/inzilico/kselection/ *Khvorykh, 2018*). Statistical phasing (i.e., phasing without a reference panel) can be error prone, but fastPHASE is commonly employed in non-model organisms and is well-suited to datasets like ours that include high density SNPs on a physical map (*Scheet and Stephens, 2006*; *Burri et al., 2015*; *Kawakami et al., 2017*).

The analysis of cross-coalescence rates requires comparisons between groups and we considered all possible combinations of groups for our analysis (*Schiffels and Wang, 2020*). In other words, we ran analyses for all 15 possible combinations (three between groups on the continent, three between populations on the islands, and nine for comparisons between the three continent groups and three island populations). For each pairwise combination, we ran the combineCrossCoal.py script from msmc-tools (https://github.com/stschiff/msmc-tools) and computed the relative cross-coalescence rate by dividing the between-populations coalescence rate by the average within-population coalescence rate. We scaled results using a mutation rate of $3 \times 10^{-9}$/gen/site and a generation time of 2 years (*Nadachowska-Brzyska et al., 2016*; *Nadachowska-Brzyska et al., 2015*).

## hapFLK

hapFLK is a tree-based method that is used to identify genomic regions that are under selection. This program permits the inclusion of two or more populations and accounts for both drift within populations (different $N_e$) and covariance across them (hierarchical structuring). We used the vcf from VQSR as input for this analysis, applying two additional filters for the inclusion of variants: minimum number of individuals/phenotype = 5 and minor allele frequency of 0.05. hapFLK also requires an estimate of the number of clusters into which haplotypes can be grouped. We ran this analysis for the complete dataset including all populations, and for a restricted dataset including only medium-distance migrants. We determined the number of clusters for each dataset separately using fastPHASE (*Scheet and Stephens, 2006*) and the cross-validation procedure mentioned earlier.

Once hapFLK is estimated, it is normalized using rlm in R, and p-values are computed from the chi-squared distribution. We used a permutation analysis to establish a threshold, beyond which genomic regions would be considered to be experiencing positive selection. Specifically, we randomly shuffled population labels 100 times, re-estimated hapFLK and p-values, recorded the lowest p-value for each randomization and set the threshold to the fifth percentile across randomizations. Once these regions were identified, we determined which population was experiencing selection by comparing branch lengths for a tree built using data from the entire genome and one built using data from the region under selection. Note that results from analyses using medium-distance migrants are plotted using the resident phenotype for illustrative purposes, but the analysis was not run using these birds.

We include birds from three resident continent populations – Cazalla de la Sierra and Gibraltar in the Iberian Peninsula along with Asni in Morocco (only three birds were sampled from this African population, precluding its use in the present analysis; *Supplementary file 4*). The Iberian Peninsula where the other two populations are found is highly heterogeneous as a result of the effects of mountains and plateaus that create variation in seasonality and, consequently, in the intensity of blackcap migratory behaviour (*Pérez-Tris and Tellería, 2002*; *Tellería et al., 2001*). There is also some evidence in our PCA to show that this heterogeneity has led to some differentiation between populations, as birds from Cazalla de la Sierra exhibit values more similar to migrants on PC2 (*Figure 1c*). Accordingly, to avoid any confounding effects from population structure, we limited our analysis to birds from Gibraltar. Results using Cazalla de la Sierra instead were very similar. For example, all of the genomic regions identified in *Table 1b* were also in the top 1% of the ΔPBS distribution when Cazalla de la Sierra was used as the continental reference population instead of Gibraltar.

## CAVIAR

The principle described above for hapFLK focusing on haplotype clusters can also be applied to SNPs (FLK). We used results from an analysis with FLK and limited to genomic regions, which showed evidence of positive selection from hapFLK, to identify independent strongly associated SNPs with CAVIAR (CAusal Variants Identification in Associated Regions [*Hormozdiari et al., 2014*]). CAVIAR was originally designed to identify independent causal SNPs in GWAS studies. We followed methods described in *Rochus et al. (2018)* to modify this method for FLK, identifying SNPs with p-values <0.0001 in hapFLK outlier regions and using a correlation matrix generated by FLK by decomposing signals into loading on orthogonal components (vs. p-values from a GWAS and LD as is traditionally done with CAVIAR).

## Δ PBS

We used a modified version of PBS (Population Branch Statistic) to complement results from hapFLK. PBS is similar to $F_{ST}$, but can include more than two populations and identifies regions within each population that exhibit differences in allele frequencies. This statistic was originally designed for three populations, but can be expanded to include more populations (*Zhan et al., 2014*). We used GL from ANGSD to obtain estimates of $F_{ST}$ following the procedure described above (summarized into windows of 2500 kb) and used the equation below to estimate PBS from these values. This equation is an example that was applied to resident populations (R), where T is log transformed $F_{ST}$ between the populations indicated in exponents:

$$\frac{T^{R-NW} + T^{R-SW} + T^{R-SE} - T^{NW-SW} - T^{SW-SE}}{3}$$

Recent papers have noted that $F_{ST}$ can be elevated by reductions in within-population variation alone and that there are many factors that can reduce variation within populations, including linked selection in areas of reduced recombination that may result from purifying selection (background selection, [*Cruickshank and Hahn, 2014*; *Noor and Bennett, 2009*]). It is unlikely that this process affects our results because recombination rate should elevate estimates of PBS in all populations, but this is not the case (*Figure 5a*). Regardless, we followed methods from *Vijay et al. (2017)* to reduce any effects that linked selection may have on our results. *Vijay et al. (2017)* used estimates of $F_{ST}$ between allopatric populations of crows that did not differ in their trait of interest to control

for the effects of linked selection, estimating the difference in estimates of $F_{ST}$ in focal populations and maximum $F_{ST}$ in non-focal allopatric populations (ΔFST). $F_{ST}$ in focal populations would have to extend beyond that in non-focal populations to be considered important in generating the trait of interest. We used the same approach for PBS. For example, ΔPBS for resident continent populations was estimated by finding the difference between PBS in residents and maximum PBS in medium-, short- and long-distance migrants.

### nSL

The former analyses (hapFLK and PBS) rely on comparisons between phenotypes. In this last analysis, we focus on the affects that selection can have within a population instead. Specifically, selective sweeps can reduce variation at both the locus under selection and its neighbours (*Smith and Haigh, 1974*). Local reductions in variation result in the presence of extended regions of haplotype homozygosity within phenotypes (long haplotypes at high frequency). nSL (number of segregating sites by length) (*Ferrer-Admetlla et al., 2014*) is similar to the more common iHS, but instead of measuring the decay of haplotype identity as a function of recombination distance, it quantifies this decay of how many mutations remain in other haplotypes present in the dataset. In this way, nSL does not require a genetic map and is more robust to variation in not only recombination rate but also mutation rate.

For this analysis, we used selscan (v.1.20a https://github.com/szpiech/selscan) and the same vcf used in hapFLK, but split by phenotype (and scaffold). We ran the data through fastPHASE first to phase haplotypes (using 50 iterations of the EM algorithm, sampling 100 haplotypes from the posterior distribution and using same number of clusters identified for hapFLK). We normalized estimates of nSL into the same 2500-kb windows used for PBS.

### Regulatory variants

Two sets of preliminary analyses were used to identify regulatory SNPs in the regions identified by hapFLK and PBS as being under selection. First, we focused on 3′ UTRs, downloading predicted mRNAs from Ensembl and NCBI for several bird species, including the Atlantic canary, White-throated sparrow, American crow, Great tit, Collared flycatcher, Zebra finch, Wild turkey, White-rumped munia, Hooded crow, Blue tit and Ground tit. We aligned these sequences with our annotation for the blackcap, and with transcripts assembled from RNAseq data obtained from the testes of a single male blackcap, to determine whether any of the strongly associated SNPs identified by CAV-IAR were within 3′ UTRs. Alignment files are available upon request.

In a second set of analyses, we used HOMER (*Heinz et al., 2010*) to identify known transcription factor binding sites (TFBS) in genomic regions under selection. Specifically, we used findMotifsGenome.pl with default settings to identify known motifs in each region and scanMotifGenomeWide.pl to identify the specific location in each region where the motif could be found (permitting no mismatches). HOMER includes known motifs for thousands of transcription factors (mostly for model organisms); we chose to focus on candidate transcription factors identified by previous studies as having an association with migration (*Ruegg et al., 2014*).

### GWAS

In our final analysis on migratory distance, we limited our dataset to short-, medium- and long-distance migrants. We coded distance phenotype as an ordinal variable from 1 to 3 and conducted a GWAS analysis using a Bayesian sparse linear mixed model (BSLMM) (*Zhou et al., 2013*). We chose BSLMM models here (instead of hapFLK) because they allow the inclusion of ordinal variable (vs. categorical with hapFLK). BSLMM models include the phenotype as the response variable and allele frequencies at a set of SNPs as the predictor variable. They also include a term for factors that influence the phenotype and are correlated with genotype (e.g., population structure). BSLMMs are adaptive models that include linear mixed models (LMM) and Bayesian variable selection regression (BVSR) as special cases and that learn the genetic architecture from the data. We ran four independent chains for each BSLMM, with a burnin of 5 million steps and a subsequent 20 million MCMC steps (sampling every 1000 steps). We report one hyperparameter from this model (PVE: the proportion of variance in phenotypes explained by all SNPs, also called chip heritability) and consider SNPs with inclusion probabilities >0.01 following *Gompert et al. (2013)*. Note, we chose to run this

analysis with GEMMA instead of hapFLK as we did with our other migratory traits (orientation and propensity). This is because our focal variable here (distance) is ordinal in nature and this fact would have been lost in hapFLK. We could not code this variable as continuous because the average distance individuals in each population travel on migration is not exactly known.

## Acknowledgements

We acknowledge funding from the Max Planck Society (MPRG grant to ML), NSERC (PDF to KD) and Regional Government of Asturias (GRUPIN to JCI, Ref.: IDI/2018/000151). We thank: Staffan Bensch, Andreas Helbig, Stuart Bearhop, and Thord Fransson for providing samples; Conny Burghardt, Heinke Buhtz and Sven Künzel for lab work; Diethard Tautz, Tobias Kaiser and Sandra Bouwhuis for comments on early drafts; Julien Dutheil, Reto Burri, Kristian Ullrich, Christine Merlin and Aldrin Lugena for discussions of analyses; and Saki Chan at BioNano for the hybrid assembly of our genome. We would also like to thank Elizabeth Scordato, Patricia Wittcopp and three anonymous reviewers for constructive feedback on an earlier version of this manuscript. Permits were provided to JCI for samples collected in Morocco (Haut Commissariat aux Eaux et Forets et a la Lutte Contre la Desertification, 206/2011, 13 Jan 2011), Cape Verde (Ministerio do Ambiente - Habitacao e Ordenamento do Territorio, 18/CITES/DNA, 17 Dec 2015) and the Azores (Instituto da Conservacao da Natureza e da Biodiversidade, 171/2008, 31 Mar 2009). JP-T received permits for samples collected in Gibraltar and Cazalla de la Sierra (Consejeria de Medio Ambiente, 50.725.548-Z, 12 May 2011), Alava (Arabako Foru Aldundia, 50.725.548-Z, 12 Apr 2011) and Guadaramma (Consejeria de Medio Ambiente - Vivenda y Ordenacion del Territorio, 10/160876.9/10, 12 Apr 2010). Thord Fransson received permits for samples collected in Stockholm (Stockholms djurförsöksetiska nämnd Dnr N 16/16 2016-02-25). Permits were provided to GS for samples collected in the remaining locations (Regierungspräsidium Freiburg, 55–8853.17/0).

## Additional information

### Funding

| Funder | Grant reference number | Author |
| --- | --- | --- |
| Natural Sciences and Engineering Research Council of Canada | PDF | Kira Delmore |
| Max-Planck-Gesellschaft | MPRG | Miriam Liedvogel |
| Regional Government of Asturias | GRUPIN: IDI/2018/000151 | Juan Carlos Illera |

The funders had no role in study design, data collection and interpretation, or the decision to submit the work for publication.

### Author contributions

Kira Delmore, Conceptualization, Resources, Data curation, Formal analysis, Supervision, Funding acquisition, Validation, Investigation, Visualization, Methodology, Writing - original draft, Project administration, Writing - review and editing; Juan Carlos Illera, Javier Pérez-Tris, Resources, Data curation, Funding acquisition, Methodology, Writing - review and editing; Gernot Segelbacher, Resources, Data curation, Funding acquisition, Writing - review and editing; Juan S Lugo Ramos, Gillian Durieux, Jun Ishigohoka, Formal analysis, Writing - review and editing; Miriam Liedvogel, Conceptualization, Resources, Data curation, Supervision, Funding acquisition, Methodology, Project administration, Writing - review and editing

### Author ORCIDs

Kira Delmore https://orcid.org/0000-0003-4108-9729
Juan Carlos Illera http://orcid.org/0000-0002-4389-0264
Jun Ishigohoka https://orcid.org/0000-0002-5713-9391
Miriam Liedvogel https://orcid.org/0000-0002-8372-8560

## Ethics

Animal experimentation: Permits to JCI for samples collected in Morocco (Haut Commissariat aux Eaux et Forets et a la Lutte Contre la Desertification, 206/2011, 13 Jan 2011), Cape Verde (Ministerio do Ambiente - Habitacao e Ordenamento do Territorio, 18/CITES/DNA, 17 Dec 2015) and the Azores (Instituto da Conservacao da Natureza e da Biodiversidade, 171/2008, 31 Mar 2009); Permits to JP-T for samples collected in Gibraltar and Cazalla de la Sierra (Consejeria de Medio Ambiente, 50.725.548-Z, 12 May 2011), Alava (Arabako Foru Aldundia, 50.725.548-Z, 12 Apr 2011) and Guadaramma (Consejeria de Medio Ambiente - Vivenda y Ordenacion del Territorio, 10/160876.9/10, 12 Apr 2010); Permit to Thord Fransson for samples collected in Stockholm (Stockholms djurförsöksetiska nämnd Dnr N 16/16 2016-02-25); Permits to GS for samples collected in the remaining locations (Regierungspräsidium Freiburg, 55-8853.17/0).

## Decision letter and Author response

Decision letter https://doi.org/10.7554/eLife.54462.sa1
Author response https://doi.org/10.7554/eLife.54462.sa2

# Additional files

## Supplementary files

• Supplementary file 1. Summary of sequencing data used for ALLPATHS-LG assembly. Libraries designated a and b are from the same library preparation but sequenced on two separate lanes.

• Supplementary file 2. Assembly statistics at each stage. The second ALLPATHS assembly follows the removal of duplicates and contaminants along with gap filling.

• Supplementary file 3. Results from satsuma showing which flycatcher chromosome each scaffold in the blackcap reference genome hit. Mean position and orientation refer to the location and orientation of scaffolds on the flycatcher genome. The last six scaffolds did not hit any of the flycatcher chromosomes. Comparing the annotation of the blackcap and zebra finch genomes suggests they match the indicated chromosomes.

• Supplementary file 4. Samples used in the present study, including their locations and details on how phenotypes were determined.

• Supplementary file 5. Extension of *Table 1* showing regions identified by hapFLK as being under selection but including the number of causal SNPs identified by CAVIAR and their location within genes. Estimates of nSL are shown (bolded if in top 1% of values, bolded and italicised if in the top 5%).

• Supplementary file 6. Additional details on genome assembly and annotation.

• Transparent reporting form

## Data availability

Sequencing data has been deposited under NCBI BioProject PRJNA616371. All other data are included in the manuscript and supporting files.

The following datasets were generated:

| Author(s) | Year | Dataset title | Dataset URL | Database and Identifier |
|---|---|---|---|---|
| Delmore K, Illera JC, Pérez-Tris J, Segelbacher G, Lugo Ramos JS, Durieux G, Ishigohoka J, Liedvogel M | 2020 | European blackcap resequencing | https://www.ncbi.nlm.nih.gov/bioproject/PRJNA616371/ | NCBI BioProject, PRJNA616371 |
| B10K Consortium | 2019 | Bird 10,000 Genomes (B10K) Project - Family phase | https://www.ncbi.nlm.nih.gov/bioproject/?term=PRJNA545868 | NCBI BioProject, PRJNA545868 |

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
