## [Decision Letter]

[Editors’ note: the authors submitted for reconsideration following the decision after peer review. What follows is the decision letter after the first round of review.]

Thank you for submitting your work entitled "The evolutionary history and genomics of European blackcap migration" for consideration by *eLife*. Your article has been reviewed by three peer reviewers, one of whom is a member of our Board of Reviewing Editors, and the evaluation has been overseen by a Senior Editor. The reviewers have opted to remain anonymous.

This manuscript compiles an enormous dataset to address questions about the genetic basis of migratory behavior in a classic model system in the field. The reviewers and reviewing editor were enthusiastic about the topic and united in finding the dataset extremely impressive. They also thought several key results of the paper were important. In particular, the data add to growing evidence that there is not a common genetic mechanism underlying migratory behavior. Furthermore, the attempt to decompose a complex behavior into component parts (propensity, distance, and orientation) was novel and interesting.

Despite this enthusiasm for the work, a number of significant issues were raised about the current presentation of the data and some of the analyses. These major concerns are synthesized below. After extensive discussion, the revisions required for the paper to be acceptable were deemed to be too substantial to be completed in the two month period allowed for revisions at *eLife*. We have therefore recommended rejection of this manuscript in its current state. Given the great promise of the dataset, however, the reviewers encourage submission of a significantly revised manuscript as a new submission in the future.

1) A primary issue identified by all reviewers is that the paper was generally hard to follow and contained many unrelated analyses. The Introduction does not set up clear questions and gives little background information on why the blackcap system is uniquely well-suited for addressing questions about the genetic basis of migratory behavior. The Introduction also does not set up the key findings of the paper. It is recommended that the authors re-focus the overall framing of the paper to highlight their key results, remove tangential analyses (see below for specifics), and generally streamline and contextualize results within the existing literature throughout the entire paper.

2) There was concern from all reviewers about the phylogeography section. These analyses were not clearly integrated with the rest the paper. Furthermore, the way the PCA and ADMIXTURE plots were used to interpret ancestral state seemed very ad-hoc, and the maximum likelihood tree was not well supported. The inference that migration is the ancestral state and was lost on islands is generally reasonable, as that is a pattern found in other taxa. However, if the authors choose to include the phylogeography section moving forward, removal of the PCA and ADMIXTURE interpretations is recommended. The authors should also acknowledge the limitations of their inference more openly, given the weak support in their data. Further, the overall contribution of the phylogeographic analysis to the main conclusions of the paper should be clarified throughout the text.

3) The use of SMC++ to infer divergence times was considered problematic, as this method assumes populations do not exchange genes after divergence. To more accurately date divergence times, the authors should use a demographic modeling approach based on the joint site frequency spectrum (e.g. dadi or moments) or evaluate alternative models with and without migration in an ABC framework. Alternatively, the authors could refocus that section on variable demography rather than the timing of splits. Demographic trajectories diverging earlier than previously estimated split times could then be a "suggestive" point, but shouldn't be the primary finding of that analysis. It might be useful to refer to Mazet et al., 2016 (https://www.ncbi.nlm.nih.gov/pmc/articles/PMC4806692/) for guidance with these interpretations.

4) There were concerns about the authors overinterpreting causal associations between migratory behavior and loci putatively under selection in both the orientation and residency sections. Particularly in the transition to residency section, outliers cannot be separated from population structure, and could therefore be due to a variety of factors unrelated to migration. The authors suggest that finding an elevated region on super scaffold 99 in both island and resident populations indicates this region may be associated with the transition to residency. This analysis is rather unconvincing, as these shared elevated regions could be due to shared ancestry. It is also not clear if the extensive subsequent discussion of candidate loci focuses only on this shared elevated region on scaffold 99, or looks at all outliers. The language in this section should tempered, alternative explanations discussed, and it should be made clear which populations are included when identifying candidate loci.

5) Analyses of selection on standing genetic variation: as with the inferences about phylogeography, these sections read as somewhat ad-hoc and did not use previously validated methods. The first section about standing genetic variation (dealing with short distance migrants) is weakly supported- removal this section is recommended. The second section, if retained, should use methods that have been previously shown through theory or simulation to distinguish selection on standing genetic variation. It is also recommended that the authors articulate clear predictions about expected results if migratory behavior arises from selection on novel vs. standing variation.

6) More information needs to be given about sampling and phenotyping. This is particularly important for explaining how phenotypes were assigned to "short-distance SW migrants," which fall into two different clusters on PC2. This is a major gap in the paper.

7) There were concerns about the analysis of migratory distance, both in classifying distance as a categorical trait and in why a different method for identifying outliers was used in this section. The analysis of distance, if retained, should either apply the same approaches for outlier detections as were used for the analyses of propensity and orientation, or clearly explain use of a different approach. There should also be further justification for classifying distance as a categorical variable.

Reviewer #1:

This paper uses the European blackcap, a classic model system in migration research, to reconstruct the evolution of migratory behavior and identify genomic regions associated with different components of migratory phenotype. To me, the most impactful results are that the authors do not find candidate loci associated with migratory behavior in other systems to be under selection in the blackcap, adding to growing evidence against a common underlying genetic architecture of migratory behavior. Also important is that the authors looked at multiple components of migratory behavior (propensity, orientation, and distance). However, while the dataset and analyses are very impressive, I think that the authors try to do too much at the expense of clarity and a focused message. Several sections of results are unclear, key information about sampling is missing, there are many tangential and speculative sections, and the paper as a whole does not have a strong unifying framework or question.

1) While the dataset here is remarkable, the structure of the paper is quite challenging. The Introduction, in my opinion, does not articulate a clear overarching question, and the questions that are laid out in the Introduction are not linked well to the analyses presented in the paper. For example, the Introduction seems to set up a study focused on the genetic architecture of migration. However, the first part of the paper is a lengthy analysis of population structure and phylogeography that seems unrelated to any topics introduced in the Introduction and is not connected to the background provided on the system. When we get to the genetic architecture component of the paper, it is not contextualized well with previous knowledge of the system, and it is therefore hard to evaluate the significance of the results. It is also not clear how the earlier pop structure and phylogeography component is related to analyses of selection and genetic architecture.

2) Very little information is given about the sampling or how phenotypes were identified. The only information provided is in the legend of Figure 1, which states that samples were collected on the breeding grounds with the exception of a few collected during the winter or during migration. This is problematic, given that the paper relies on individual-level phenotype assignments for these samples in all analyses. Explanations of how migratory distance, propensity, and orientation are determined for individual samples are needed.

3) The section on inferring ancestral state reads as very ad-hoc and confusing. The way the PCA is interpreted is not convincing and is not supported by any citations. The results from TREEMIX and ADMIXTURE are conflicting, but no explanation for how to interpret this discordance is offered. The purpose of this section overall is also not entirely clear- how do these analyses link back to overall questions about genetic architecture of migration?

4) The treatment of short-distance migrants is confusing. Based on the PCA, ADMIXTURE plot, and the map, it looks to me like these populations are admixed between residents and migrants (possibly due to post-Pleistocene secondary contact, based on the authors' explanation of biogeography). It is unclear to me how any of these data support variation in migratory behavior arising from standing genetic variation. Are the authors suggesting that individuals in the short-distance migrant population have different migratory strategies (e.g. some are residents and some are migrants) due to variation within the population? Later evidence for the role of standing genetic variation is more convincing. The purpose of this section needs to be made clearer.

5) I was surprised, given low support for the maximum likelihood tree and the history of gene flow indicated by TREEMIX, that the authors didn't use a method of demographic reconstruction that allows migration and determination of the timing of admixture events.

6) The way results on the propensity to migrate are presented strongly suggests that regions under selection in non-migratory populations are directly associated with loss of migratory behavior. However, these regions could just as reasonably be under other sources of divergent selection between these populations (e.g. ecological differences, differences in diet, etc). Couldn't shared regions under selection in island and resident populations also be due to gene flow and shared ancestry in these populations, as indicated by TREEMIX? Overall, I don't find it very convincing that regions under selection in non-migratory populations are directly associated with migratory propensity. The authors should either temper this section and discuss alternative explanations, or make the support for their assertion clearer.

7) In the first analyses of genomic regions under selection, the authors "consider all three traits together." What does this mean? What groups of samples were actually compared?

8) I wondered why a different method was used for the analysis of migratory distance than was used for orientation and propensity. Why not make these analyses comparable? This last section in general felt somewhat tacked-on and not well integrated into the rest of the paper.

9) Overall, I think the results could be more clearly contextualized within the broader literature throughout the paper. The authors have a very cool dataset, but the parts that are novel and exciting are not highlighted very well throughout. The amount of background information given seems to assume familiarity with the blackcap system, which limits the accessibility of the paper to a broad audience. I think more effort needs to be put towards explaining why this system is so well-suited for asking these questions, and what new things we learn about migratory behavior from these analyses.

Reviewer #2:

In this manuscript, the authors use whole-genome data to study the genetic basis of three migratory traits in European blackcaps: the propensity to migrate, distance, and orientation. Leveraging the diversity of migratory strategies in blackcaps, the authors document associations between migratory behavior and SNPs in regulatory regions – providing the first genome-wide characterization of migratory behavior in this species. Not only is this an interesting study system, but the sampling and experimental design also provide a robust foundation for investigating the genomic basis of migratory behavior. In addition, I found the methodological approach to be sound and thorough and was particularly impressed with the sample sizes/coverage for re-sequencing data as well as the quality of the reference genome. Overall, I thought it was an interesting data set and enjoyed reading the manuscript.

I have a few comments/questions outlined below. It is my opinion that, if these concerns are addressed, this work would make a nice contribution to the existing literature.

Subsection “Resident populations evolved from a migratory ancestor”: While I like this section, I think you can combine the TREEMIX and demographic analyses to cover most of this section in a more compelling way. While I don't entirely disagree, I find the use of the PCA and ADMIXTURE results to infer ancestral state a bit weak compared to the demographic data. I think you have enough analyses outside of the PCA and ADMIXTURE results to draw conclusions regarding ancestral state that are more appropriate for this type of question. I would recommend that you narrow this section down to focus on fewer, but more definitive, analyses, as it would strengthen the overall argument.

Subsection “Evidence for standing genetic variation in short distance migrants”: Similar to my comment above, I found parts of this section distracting and less compelling compared to the rest of the manuscript. I think you could remove this section as I am not really sure it adds much to the paper. I feel similarly about subsection “Selection on shared genetic variation could facilitate rapid changes in migration”.

Subsection “Resequencing analysis” paragraph three: This is more just curiosity: did you try using HaplotypeCaller from GATK? I think the fact that variant calling in GATK is combined with other programs makes this a robust approach but was wondering why you chose UnifiedGenotyper.

Subsection “Linkage disequilibrium”: I don't believe that there is any mention of linkage disequilibrium in the main text of the manuscript. May be worth mentioning somewhere in the results

Figure 4: Add a Y-axis label to panel c

Figure 5: I think the figure is very aesthetically pleasing, but I am really struggling to understand what is going on in this figure. It may just be a matter of clarifying the legend, or perhaps it is just me. But I am not convinced this figure adds much. Also, I think you mean to say "panel f shows haplotype.…."

Reviewer #3:

In this study the authors analyze genetic variation in what has historically been the model species for songbird migration and present a set of results covering both phylogeographic history and selection on genes putatively associated with migratory behaviors. Helbig and Berthold's captive breeding studies in black-caps showing that migratory orientation is heritable are classics in ornithology, and an obvious followup question is "what genes are responsible for this behavior?", so I was excited to see a rigorous genome-wide analysis of the species. From that perspective I think one of the most interesting thing here is that this is now the third study (fourth if you count the brand new Toews et al. PNAS study) looking for associations of specific genomic regions with migratory behavior, and as far as I can tell there are zero overlapping outlier regions across these studies. The authors make this point but I think it could be emphasized, particularly given that much of the interest in migration as a tractable trait for genetic mapping is driven by the captive breeding studies conducted in this species.

In general I thought the new data in this paper was very good, but the results presented cover so much territory that individual analyses sometimes feel rushed and the whole picture is hard to follow. I have a few methodological issues with the phylogeography, and for the selection/association analyses I think the more speculative post-hoc analyses of specific genes should be limited. One general issue that should be addressed more directly when describing results for regions putatively associated with migratory behaviors is that migration itself may be tangential to the actual selective force – climatic variation on either the breeding or wintering range, dietary differences across geographic regions, or any other environmental factor varying among populations could create the basis for selection that could be detected by approaches like PBS. Because migratory phenotypes covary with many of these other environmental factors, it is inevitably going to be difficult to identify genes that drive (rather than being driven by) aspects of migration like orientation, distance, or phenology.

That being said I think this is an important paper in the area of migration genomics because it is in the only system with truly compelling captive-breeding results from crosses and because the dataset is excellent.

Subsection “Differentiation between populations is low”: Are migrant populations differentiable on other PC's? What about if PCA is run on just continental, or just migrant populations alone? Because PCA will represent variance in the full dataset, running it with divergent island populations may mask differentiation among migrant populations. Probably same issues in the admixture analysis. I'd like to see a supplemental figure showing at least PC1-2 when the analysis is run on just continental birds.

Subsection “Resident populations evolved from a migratory ancestor”: The methods for ancestral state reconstruction here were unfamiliar to me and seemed ad-hoc. Are there citations available for studies outlining the logic of using PCA and/or admixture results for this task? As it is the clearest evidence seemed to be from the phylogenetic analysis, which relies on poorly supported nodes in a topology constructed using a method that effectively assumes no gene flow.

Subsection “Divergence began ~250,000 year ago”: SMC++ assumes populations are isolated, so it isn't an appropriate method for dating population splits in groups that continue to exchange genes after divergence. In addition, the mutation rate and generation times (though they look about right relative to other migratory birds) don't appear to be pulled from the citation listed (Noor and Bennett, 2009 – apologies if I missed it buried in there somewhere), and these will directly scale the inferred timing of population size changes. The message that divergence is old is likely right – if gene flow occurs but isn't being captured by the analysis then divergence times should be *even older*, but if divergence time estimation is the goal then a demographic inference method explicitly incorporating gene flow and returning estimates of split times should be employed, and the empirical rates should be better justified. I'd suggest demographic modeling of the joint site frequency spectrum in dadi or moments. In addition, the end of this section regarding the possible speed of evolution of variation in migratory traits should acknowledge that much of that literature is based on the documented contemporary evolution of the NW migration and not on biogeographic reconstruction of island colonizations.

Subsection “The transition to residency may be controlled by regulatory elements” paragraph two: It was hard to tell what in this paragraph referred to results from this study, vs results from previous studies. Are Clock/Npas2/bmal1 in an outlier region here? Or is it just that there is one bHLH motif found in an outlier region? If the latter, I'd suggest cutting the second half of this section as the evidence of any specific regulatory element being involved seems quite weak.

Subsection “Examining genes associated with the transition to residency” paragraph one: Needs citations.

Subsection “Limited overlap with previous genomic studies examining this trait” paragraph two: Worth making the point here that though this gene is *associated* with migratory orientation, the proposed mechanism is not at all *causal* for orientation. This point could be more generally made in the Introduction, as well.

Subsection “Considerable variance in migratory distance is controlled by genetic variation”: This analysis did not seem particularly compelling, and (given that migratory distance is actually a continuous trait) the binning of distance as an ordinal trait seems likely to offer little power to identify causal alleles in a small cohort like this. It also isn't clear to me why a new method is used here that wasn't used in the previous association/selection analyses. I suggest cutting this section.

Subsection “Selection on shared genetic variation could facilitate rapid changes in migration”: I like this question quite a bit and think there are some cool results here, but I found this section hard to follow. I think it needs a rewrite, and the methods should be better justified. I suggest starting with a paragraph laying out expectations for what you expect given selection from novel vs standing variation (see Barret and Schluter, 2008), and using only analyses that have been previously proposed and shown through either theory or simulation to distinguish these processes.

Subsection “GWAS.”: What specifically was the factor used to represent population structure here? A matrix of pairwise genetic distance over the whole genome? Please provide a little more detail on your implementation of this method.

Figure 1C: These colors seem to partially but not entirely match the map, which is confusing (especially when referring back to this figure for color references). I'd use different colors than in the map. Greyscale would work ok with K=3.

Figure 5 legend: the letters seem to be off here – please check that e and f are correctly referenced, and discuss panels in order (a-e).

[Editors’ note: further revisions were suggested prior to acceptance, as described below.]

Thank you for resubmitting your work entitled "The evolutionary history and genomics of European blackcap migration" for further consideration by *eLife*. Your revised article has been evaluated by Patricia Wittkopp (Senior Editor) and a Reviewing Editor.

The manuscript has been improved but there are some remaining issues that need to be addressed before acceptance, as outlined below:

All three reviewers found the manuscript much improved. Only reviewer 1 had a substantial outstanding comment that requires clarification. After consultation, the other two reviewers and the Reviewing Editor concur with the points raised by reviewer 1. Please address the below questions about how individuals were assigned to different phenotype groups in the different analyses. The manuscript is provisionally recommended for acceptance pending satisfactory revision on these points. The authors are also encouraged to consider the minor points raised by all three reviewers.

Reviewer #1:

I find the resubmitted manuscript to be a substantial improvement over the initial submission. The authors are to be commended for their thorough revision and for incorporating reviewer comments.

The only major comment from my previous review that I feel is not clarified in the revision is the way the three different migratory phenotypes are combined in the demographic analyses and the initial hapFLK/PBS/nSL analyses that include all populations (point #7 from my review). Let me better explain my confusion: in the analyses of population structure (Figure 1 Fst plot), there are nine populations with distinct phenotypes: continental residents, short distance sw migrants, long distance SE migrants, medium distance SW, SE, and NE migrants, and the three island populations. However, in subsequent analyses, these populations are grouped together in different ways that are not clearly explained. For the demographic analysis, the medium- and long- distance migrants are grouped together, despite variation in orientation among these populations. Why is this? Likewise, in the analyses of the genetic basis of traits, the authors describe analyzing all three phenotypes (orientation, distance, and propensity); however, each of these of course has multiple categories (e.g. SW, SE, and NE orientation). In the legend for Figure 3, they note that they are grouping birds into 5 phenotypes for panels A and B and 3 phenotypes for panels B and C. Figure 3—figure supplement 1 shows six different phenotypes (7 including those shown in Figure 3) for the ΔPBS analysis. The authors note that islands were excluded and residents limited to a single population, but it is otherwise unclear to me what these different phenotype groups are. Adding to the confusion is that 1) the analysis of differentiation and selection comes after the demographic modeling, which did not consider orientation in the grouping of phenotypes; 2) Figure 5 also combines the medium and long- distance migrants and removes orientation; and 3) sample sizes are not given for each population used in the comparisons. Some clarification of exactly which populations and phenotypes are being compared in the analyses, throughout this this section is needed, as well as justification for why medium- and long- distance migrants are combined for some analyses but not others, and clear reporting of sample sizes for each comparison in the main text (please remove from legend in Figure 3).

Reviewer #2:

The authors have done a commendable job revising this manuscript. The Introduction is much easier to follow and sets up the study nicely. The revised Materials and methods and increased focus on linking methods/results directly to specific study questions and objectives greatly improve the overall flow and readability of the paper. I appreciate the effort that has gone into addressing reviewer comments.

Reviewer #3:

The authors have done a really excellent job with this revision and I'm pleased to recommend acceptance. The revised demographic analysis and Introduction are significant improvements and all of my major questions from the first submission have been addressed well in the response.

---

## [Author Response]

[Editors’ note: the authors resubmitted a revised version of the paper for consideration. What follows is the authors’ response to the first round of review.]

1) A primary issue identified by all reviewers is that the paper was generally hard to follow and contained many unrelated analyses. The Introduction does not set up clear questions and gives little background information on why the blackcap system is uniquely well-suited for addressing questions about the genetic basis of migratory behavior. The Introduction also does not set up the key findings of the paper. It is recommended that the authors re-focus the overall framing of the paper to highlight their key results, remove tangential analyses (see below for specifics), and generally streamline and contextualize results within the existing literature throughout the entire paper.

Thank you for summarizing the reviewers concerns here. We have taken them on board, re-writing the Introduction entirely and reducing the phylogeographic portion of our study substantially, including all discussion of ancestral state derived from our results (see our response to your second comment).

Our new Introduction highlights the importance of phylogeographic studies with blackcaps. For example, these studies provided important evolutionary context for work on the genetics of migration. Limited genetic differentiation was documented between blackcap populations suggesting transitions between migratory states must be very rapid and may involve on a small number of genetic changes. These studies also speak to the timing and number of transitions that have occurred within blackcaps between migratory and resident behaviour. This information is important for interpreting local signatures of differentiation.

All of the phylogeographic work conducted prior to our study relied on limited mitochondrial sequence data making our work – updating results with genome-wide data – essential. Our results support previous findings and uncover novel patterns that are important for interpreting our findings on the genetics of migration. For example, we document genomic differentiation between migrants and both resident continent and island populations. This had not been observed before and guided our analysis on the genetics of migratory orientation specifically as we knew the inclusion of genomically differentiated resident birds was limiting our power to detect signatures related to this trait. The exclusion of residents allowed us to identify regions under positive selection in the NW phenotype, one of the most exciting aspects of the blackcap system that we have been able to underline with our results.

2) There was concern from all reviewers about the phylogeography section. These analyses were not clearly integrated with the rest the paper. Furthermore, the way the PCA and ADMIXTURE plots were used to interpret ancestral state seemed very ad-hoc, and the maximum likelihood tree was not well supported. The inference that migration is the ancestral state and was lost on islands is generally reasonable, as that is a pattern found in other taxa. However, if the authors choose to include the phylogeography section moving forward, removal of the PCA and ADMIXTURE interpretations is recommended. The authors should also acknowledge the limitations of their inference more openly, given the weak support in their data. Further, the overall contribution of the phylogeographic analysis to the main conclusions of the paper should be clarified throughout the text.

We are confident that our new Introduction integrates our phylogeographic work much better than before. We discuss findings in the blackcap on the genetics of migration first and devote a second paragraph to phylogeographic studies, highlighting their importance for interpreting data on the former topic.

We have kept the PCA and ADMIXTURE analyses in our study but removed TREEMIX and ML trees. We do not use any of these analyses to infer ancestral states; we only use them to discuss patterns of genomic differentiation between populations and study demographic history. As we mentioned above, these analyses highlighted new results for the blackcap system that were important for setting up subsequent analyses (e.g. removing resident continent birds) and interpreting findings (e.g. given the low levels of population differentiation we documented in the phylogeographic portion of our work we can be confident population differentiation between migratory phenotypes (i.e. NW, SW and SE migrants) is not affecting our local genomic results). Combined with other observations from the system, genomic differentiation between resident populations also suggests that parallel signatures of selection are independent but make use of the same variation.

3) The use of SMC++ to infer divergence times was considered problematic, as this method assumes populations do not exchange genes after divergence. To more accurately date divergence times, the authors should use a demographic modeling approach based on the joint site frequency spectrum (e.g. dadi or moments) or evaluate alternative models with and without migration in an ABC framework. Alternatively, the authors could refocus that section on variable demography rather than the timing of splits. Demographic trajectories diverging earlier than previously estimated split times could then be a "suggestive" point, but shouldn't be the primary finding of that analysis. It might be useful to refer to Mazet et al., 2016 (https://www.ncbi.nlm.nih.gov/pmc/articles/PMC4806692/) for guidance with these interpretations.

We agree that it may have been problematic to use of SMC++ to date population divergence in our initial submission. Accordingly, in the revised version of our manuscript we used a different program for demographic inference and complimented this analysis with cross-coalescent rate analyses. The approach that we used was introduced by the MSMC model (Schiffels and Durbin, 2011) and extended in MSMC2 (Malaspinas et al., 2012). Comparing the rate of coalescence between pairs of individuals taken in two populations to the rate of coalescence within each population allows dating the differentiation of two populations. This approach is particularly well suited to our dataset as it does not require extensive sample sizes (as for instance needed by SFS estimation) and makes use of complete genome data as it intrinsically accounts for linkage disequilibrium. In addition to the dating of population differentiation, cross-coalescence rates can be used to infer the temporal pattern of migration rate between populations (https://www.biorxiv.org/content/10.1101/585265v1.full). As our aim is to date population divergence, we decided to restrict our analyses to the differentiation timing and did not compute estimates of migration rates.

4) There were concerns about the authors overinterpreting causal associations between migratory behavior and loci putatively under selection in both the orientation and residency sections. Particularly in the transition to residency section, outliers cannot be separated from population structure, and could therefore be due to a variety of factors unrelated to migration. The authors suggest that finding an elevated region on super scaffold 99 in both island and resident populations indicates this region may be associated with the transition to residency. This analysis is rather unconvincing, as these shared elevated regions could be due to shared ancestry. It is also not clear if the extensive subsequent discussion of candidate loci focuses only on this shared elevated region on scaffold 99, or looks at all outliers. The language in this section should tempered, alternative explanations discussed, and it should be made clear which populations are included when identifying candidate loci.

We have noted the potential for other selection pressures to lead to the patterns of positive selection.

We believe we have presented strong evidence for parallel positive selection on Super-Scaffold_99 related to the transition from migrants to residents. We outline our reasons below and include them in our manuscript but are of course open to further discussion on the topic.

– Three separate population genetic measures suggest this region is under positive selection: hapFLK (which relies on haplotypes and controls for hierarchical population structure), ΔPBS (which uses SNPs and controls for linked selection) and nLS (which is a measure of within population variation that relies on haplotypes). The effect linked selection can have on signatures of selection is a very prominent problem discussed in the literature and we have used ΔPBS to address this problem (proposed in Vijay et al., 2017). Population structure between populations is low to begin with and while there was some evidence for differentiation between migrants and residents, the actual level of differentiation between migrant and resident populations is low (~0.03), should have more of a genome-wide effect and is controlled for in the analysis using hapFLK.

– It is very unlikely that selection related to other ecological features is responsible for repeated patterns of positive selection on Super-Scaffold_99 in resident continent and island birds as ecological features (both biotic and abiotic) are very different between these regions. For example, considering climate, the Iberian Peninsula is quite variable but in general it would be considered mediterranean in nature (hot, dry summers and cool, wet winters). By contrast, many of the Atlantic islands have limited seasonality and are dominated by hot desert like conditions (the Canary Islands and Cape Verde). The Azores are more seasonal (temperate with no dry season and a mild/hot summer) and these islands occur across a broad latitudinal range with increasing temperatures (and decreasing precipitation) as you move towards the equator (Cropper, 2013). Community assemblages experienced by resident populations are also different, with islands having lower diversity in general but exhibiting their own variation as well. For example, the Canary Islands host ~3x the number of bird species and endemics owing to their proximity to the mainland and greater surface area (Valente et al., 2017).

– Finally, very similar findings and interpretations have been made by other authors, including Zhan et al., 2014, who documented parallel signatures of positive selection in transitions from migratory monarchs in North America to islands. We are lucky we have a system where this kind of comparison (multiple transitions) can be made, providing more support than a single comparison.

We do not see why the source of variation upon which selection acts precludes the region on Super-Scaffold_99 from being involved in the transition to residency. Whether the variation comes from new mutations, gene flow or standing genetic variation should not matter. It is still variation in this region (with its specific genes and regulatory regions) that experiences positive selection and is likely involved in the transition. The Atlantic islands were colonized by blackcaps in different waves and before areas in southern Europe (Perez-Tris et al., 2004, Dietzen et al., 2008). It seems the use of shared variation (gene flow or standing genetic variation) could have facilitated these transitions and is actually quite interesting re. how rapid phenotypic transitions can occur.

5) Analyses of selection on standing genetic variation: as with the inferences about phylogeography, these sections read as somewhat ad-hoc and did not use previously validated methods. The first section about standing genetic variation (dealing with short distance migrants) is weakly supported- removal this section is recommended. The second section, if retained, should use methods that have been previously shown through theory or simulation to distinguish selection on standing genetic variation. It is also recommended that the authors articulate clear predictions about expected results if migratory behavior arises from selection on novel vs. standing variation.

We have removed the first section on standing genetic variation and short distance migrants. Please see our response to reviewer 3 concerning the use of previous methods for inferring standing genetic variation. Briefly, we used methods outlined in the reference they suggested (Barrett and Schluter 2008) to the best of our ability; most of the methods described rely on the availability of an ancestral population (e.g. marine sticklebacks for freshwater adaptation in the three-spinned stickleback) and the authors themselves (i.e. Barrett and Schluter 2008) note there are limitations to the methods they suggest (e.g. looking for a “soft” sweep requires knowledge of what a “hard” sweep would look like in our system).

6) More information needs to be given about sampling and phenotyping. This is particularly important for explaining how phenotypes were assigned to "short-distance SW migrants," which fall into two different clusters on PC2. This is a major gap in the paper.

We had summarised all phenotyping and sampling details in Supplementary file 4, but now more prominently further added descriptions in the respective Materials and method part. Here we clarify that in most cases we scored populations, not individuals (i.e. all individuals in a population had equal phenotype even if we assume that there may be some within-population variation). This being said, our scoring methods in these locations (specifically those sampled on the Iberian Peninsula) were based on morphometry of individuals, and there is a strong correlation between wing length and migration distance in blackcaps. Specifically, the two north-Iberian populations are short winged compared to central and northern European blackcaps, indicating short-distance movements. With respect to the short-distance SW migrants we know they are migratory because they leave in winter (winter blackcap censuses in these localities resulted in zero counts in all cases). Southern residents are known to be resident because of the combination of extremely short and rounded wings and individual ringing recoveries of individuals that spend the whole year in the same territory.

7) There were concerns about the analysis of migratory distance, both in classifying distance as a categorical trait and in why a different method for identifying outliers was used in this section. The analysis of distance, if retained, should either apply the same approaches for outlier detections as were used for the analyses of propensity and orientation, or clearly explain use of a different approach. There should also be further justification for classifying distance as a categorical variable.

We have provided more context for this analysis in our response to reviewers 1 and 2. We cannot quantify migratory distance as continuous as we do not know how far an average individual from each population travels and we chose to retain some of this variation by quantifying distance as an ordinal variable. If we had used the same method as our work on migratory orientation (hapFLK) we would have lost the continuous nature of this variable all together. We have included a note in the manuscript to this effect.

Reviewer #1:1) While the dataset here is remarkable, the structure of the paper is quite challenging. The Introduction, in my opinion, does not articulate a clear overarching question, and the questions that are laid out in the intro are not linked well to the analyses presented in the paper. For example, the Introduction seems to set up a study focused on the genetic architecture of migration. However, the first part of the paper is a lengthy analysis of population structure and phylogeography that seems unrelated to any topics introduced in the Introduction and is not connected to the background provided on the system. When we get to the genetic architecture component of the paper, it is not contextualized well with previous knowledge of the system, and it is therefore hard to evaluate the significance of the results. It is also not clear how the earlier pop structure and phylogeography component is related to analyses of selection and genetic architecture.

Thank you for these suggestions. We have re-written the Introduction, providing a more equitable consideration of both migration genetics and phylogeography and highlighting how results from phlogeographic studies are relevant to our understanding of migration genetics. For example, classic studies on both migration genetics and phylogeography highlight how rapidly migratory behaviour can evolve and limited genetic differentiation between blackcap phenotypes could suggest the effect sizes of loci underlying migration are small (or that a small number of genes control these traits). Concerning the inclusion of phylogeographic analyses as a whole, we feel they provide an important setting for subsequent analyses of migration genetics. For example, in the first analyses on this topic using hapFLK and all phenotypes we find most signatures of selection are limited to residents, likely reflecting the fact these populations evolved resident phenotypes as they colonized areas further south in Europe and on the Atlantic islands. Note we have reduced the section on phylogeography substantially including the exclusion of several analyses including TREEMIX and the ML tree.

2) Very little information is given about the sampling or how phenotypes were identified. The only information provided is in the legend of Figure 1, which states that samples were collected on the breeding grounds with the exception of a few collected during the winter or during migration. This is problematic, given that the paper relies on individual-level phenotype assignments for these samples in all analyses. Explanations of how migratory distance, propensity, and orientation are determined for individual samples are needed.

We have summarised all phenotyping and sampling details in Supplementary file 4, but now more prominently further added descriptions in the respective method part, we have also fully replied to a similar comment raised by the editor, see above. There we clarify that in most cases we scored populations, not individuals (i.e. all individuals in a population had equal phenotype even if we assume that there may be some within-population variation). Scoring methods in these locations (specifically those sampled on the Iberian Peninsula) were based on morphometry of individuals, and there is a strong correlation between wing length and migration distance in blackcaps. Other measures of phenotype are based on ringing-recovery data, stable isotope assessments as detailed in Supplementary file 4.

3) The section on inferring ancestral state reads as very ad-hoc and confusing. The way the PCA is interpreted is not convincing and is not supported by any citations. The results from TREEMIX and ADMIXTURE are conflicting, but no explanation for how to interpret this discordance is offered. The purpose of this section overall is also not entirely clear- how do these analyses link back to overall questions about genetic architecture of migration?

Thank you for this comment. We have removed all discussion of ancestral state from our Results. For example, we have excluded the ML tree and discuss results from the PCA, ADMIXTIRE and FST in terms of population differentiation only. All speculation has been removed. This section has been reduced substantially in response to comments from all reviewers.

4) The treatment of short-distance migrants is confusing. Based on the PCA, ADMIXTURE plot, and the map, it looks to me like these populations are admixed between residents and migrants (possibly due to post-Pleistocene secondary contact, based on the authors' explanation of biogeography). It is unclear to me how any of these data support variation in migratory behavior arising from standing genetic variation. Are the authors suggesting that individuals in the short-distance migrant population have different migratory strategies (e.g. some are residents and some are migrants) due to variation within the population? Later evidence for the role of standing genetic variation is more convincing. The purpose of this section needs to be made clearer.

Thank you for noting this. We have removed this section to focus on our main questions.

5) I was surprised, given low support for the maximum likelihood tree and the history of gene flow indicated by TREEMIX, that the authors didn't use a method of demographic reconstruction that allows migration and determination of the timing of admixture events.

Thank you for noting this. In the revised version of our manuscript, we complemented our demographic inference with cross-coalescent rate analyses. The approach that we used was introduced by the MSMC model (Schiffels and Durbin, 2011) and extended in MSMC2 (Malaspinas et al., 2012). Comparing the rate of coalescence between pairs of individuals taken in two populations to the rate of coalescence within each population allows dating the differentiation of two populations. This approach is particularly well-suited to our dataset as it does not require extensive sample sizes (as for instance needed by SFS estimation) and make use of complete genome data as it intrinsically accounts for linkage disequilibrium. In addition to the dating of population differentiation, cross-coalescence rates can be used to infer the temporal pattern of migration rate between populations (https://www.biorxiv.org/content/10.1101/585265v1.full). As our aim is to date population divergence, we decided to restrict our analyses to the differentiation timing and did not compute estimates of migration rates.

6) The way results on the propensity to migrate are presented strongly suggests that regions under selection in non-migratory populations are directly associated with loss of migratory behavior. However, these regions could just as reasonably be under other sources of divergent selection between these populations (e.g. ecological differences, differences in diet, etc). Couldn't shared regions under selection in island and resident populations also be due to gene flow and shared ancestry in these populations, as indicated by TREEMIX? Overall, I don't find it very convincing that regions under selection in non-migratory populations are directly associated with migratory propensity. The authors should either temper this section and discuss alternative explanations, or make the support for their assertion clearer.

We have added a note in the manuscript that parallel signatures of divergent selection in resident populations could derive from a different source of selection than migration but do not believe this is the case as ecological features (biotic and abiotic) experienced by birds on the continent and islands are very different. For example, climate on the Iberian Peninsula is variable but in general it would be considered mediterranean in nature (hot, dry summers and cool, wet winters). By contrast, many of the Atlantic islands have limited seasonality and are dominated by hot desert like conditions (the Canary Islands and Cape Verde). The Azores are more seasonal (temperate with no dry season and a mild/hot summer) and more generally, these islands occur across a broad latitudinal range with increasing temperatures (and decreasing precipitation) as you move towards the equator, Cropper, 2013). Community assemblages experienced by resident populations are also different, with islands having lower diversity in general but exhibiting their own variation as well. For example, the Canary islands host ~3x the number of birds species and endemics owing to their proximity to the mainland and greater surface area (Valente et al., 2017). A very similar argument was made by Zhan et al., 2014, in their observation that resident monarch butterflies on islands derive from continent migrant monarchs.

Concerning the source of these parallel signatures of selection, we agree these signatures could be due to gene flow or shared ancestry. We are not sure how this affects our results or interpretations. Whether the source of variation is from new mutations or shared variation does not affect our conclusion that the same regions are under selection in resident continent and island populations. Wherever the variation comes from it’s under positive selection that we believe is related to the transition from migratory to resident in these locations following their colonization (although as we discuss above we have included a note about the role other selection pressures could play). There is strong evidence from previous phylogeographic studies that colonization of the islands and southern regions of the continent occurred at different times (and at least two colonizations of the islands have occurred [0.3-3 Myr to Canaries and 4,000-40,000 ya to the other islands], Perez-Tris et al., 2004, Dietzen et al., 2008).

7) In the first analyses of genomic regions under selection, the authors "consider all three traits together." What does this mean? What groups of samples were actually compared?

The three phenotypes are orientation, distance and propensity. This means that all birds were included except for resident island populations (n=15), and resident continent birds from Asni and Cazalla (see text, n=13). We have made this clearer.

8) I wondered why a different method was used for the analysis of migratory distance than was used for orientation and propensity. Why not make these analyses comparable? This last section in general felt somewhat tacked-on and not well integrated into the rest of the paper.

Migration distance is an ordinal variable: short (1), medium (2), and long (3) distance migrants. We did not want to lose this information by running this analysis through hapFLK where it would be considered a categorical variable. GEMMA allows you to code your phenotype as ordinal. We have made this clear in the manuscript and think this analysis is integrated well enough as it is clear in our statement of objectives we will be studying the genetic basis of all three traits and this is restated at the start of the localized analyses as well.

9) Overall, I think the results could be more clearly contextualized within the broader literature throughout the paper. The authors have a very cool dataset, but the parts that are novel and exciting are not highlighted very well throughout. The amount of background information given seems to assume familiarity with the blackcap system, which limits the accessibility of the paper to a broad audience. I think more effort needs to be put towards explaining why this system is so well-suited for asking these questions, and what new things we learn about migratory behavior from these analyses.

Thank you for this suggestion. We have re-written the Introduction to provide much more background information on the blackcap.

Reviewer #2:[…] I have a few comments/questions outlined below. It is my opinion that, if these concerns are addressed, this work would make a nice contribution to the existing literature.Subsection “Resident populations evolved from a migratory ancestor”: While I like this section, I think you can combine the TREEMIX and demographic analyses to cover most of this section in a more compelling way. While I don't entirely disagree, I find the use of the PCA and ADMIXTURE results to infer ancestral state a bit weak compared to the demographic data. I think you have enough analyses outside of the PCA and ADMIXTURE results to draw conclusions regarding ancestral state that are more appropriate for this type of question. I would recommend that you narrow this section down to focus on fewer, but more definitive, analyses, as it would strengthen the overall argument.

Thank you for this suggestion. We have eliminated all discussion of ancestral states in our analysis. We have removed the ML tree as population splits were not well supported along with TREEMIX. We have not removed our PCA or ADMIXTURE analyses. Instead we use them to describe population differentiation alone. Their use in this capacity is well supported in the literature. We have described this decision in full in our response to the editor above.

Subsection “Evidence for standing genetic variation in short distance migrants”: Similar to my comment above, I found parts of this section distracting and less compelling compared to the rest of the manuscript. I think you could remove this section as I am not really sure it adds much to the paper. I feel similarly about subsection “Selection on shared genetic variation could facilitate rapid changes in migration”.

Thank you for this suggestion. We have removed both of these sections.

Subsection “Resequencing analysis” paragraph three: This is more just curiosity: did you try using HaplotypeCaller from GATK? I think the fact that variant calling in GATK is combined with other programs makes this a robust approach but was wondering why you chose UnifiedGenotyper.

We relied mostly on genotype probabilities from ANGSD and so did not think the additional computational time required for HaplotypeCaller is necessary.

Subsection “Linkage disequilibrium”: I don't believe that there is any mention of linkage disequilibrium in the main text of the manuscript. May be worth mentioning somewhere in the results

We have removed this section, thank you for highlighting this.

Figure 4: Add a Y-axis label to panel c

Thank you for noting this. We have done so.

Figure 5: I think the figure is very aesthetically pleasing, but I am really struggling to understand what is going on in this figure. It may just be a matter of clarifying the legend, or perhaps it is just me. But I am not convinced this figure adds much. Also, I think you mean to say "panel f shows haplotype.…."

We have reduced the number of panels, split this into three figures (Figure 4, 5 and Figure 4—figure supplement 1) and improved the figure legends.

Reviewer #3:[…] In general I thought the new data in this paper was very good, but the results presented cover so much territory that individual analyses sometimes feel rushed and the whole picture is hard to follow. I have a few methodological issues with the phylogeography, and for the selection/association analyses I think the more speculative post-hoc analyses of specific genes should be limited. One general issue that should be addressed more directly when describing results for regions putatively associated with migratory behaviors is that migration itself may be tangential to the actual selective force – climatic variation on either the breeding or wintering range, dietary differences across geographic regions, or any other environmental factor varying among populations could create the basis for selection that could be detected by approaches like PBS. Because migratory phenotypes covary with many of these other environmental factors, it is inevitably going to be difficult to identify genes that drive (rather than being driven by) aspects of migration like orientation, distance, or phenology.

We have included this caveat, that other selection pressures may be responsible for signatures of positive selection in our study but note, in the case of consistent patterns exhibited by resident island and continent populations this is unlikely (see our response to editor and the revised text for a full description).

That being said I think this is an important paper in the area of migration genomics because it is in the only system with truly compelling captive-breeding results from crosses and because the dataset is excellent.Subsection “Differentiation between populations is low”: Are migrant populations differentiable on other PC's? What about if PCA is run on just continental, or just migrant populations alone? Because PCA will represent variance in the full dataset, running it with divergent island populations may mask differentiation among migrant populations. Probably same issues in the admixture analysis. I'd like to see a supplemental figure showing at least PC1-2 when the analysis is run on just continental birds.

We have added the requested figure to the supplementary material (Figure 1—figure supplement 1). There is no additional structure between migratory populations when the island populations are removed.

Subsection “Resident populations evolved from a migratory ancestor”: The methods for ancestral state reconstruction here were unfamiliar to me and seemed ad-hoc. Are there citations available for studies outlining the logic of using PCA and/or admixture results for this task? As it is the clearest evidence seemed to be from the phylogenetic analysis, which relies on poorly supported nodes in a topology constructed using a method that effectively assumes no gene flow.

We have removed all discussion on ancestral states.

Subsection “Divergence began ~250,000 year ago”: SMC++ assumes populations are isolated, so it isn't an appropriate method for dating population splits in groups that continue to exchange genes after divergence. In addition, the mutation rate and generation times (though they look about right relative to other migratory birds) don't appear to be pulled from the citation listed (Noor and Bennett, 2009 – apologies if I missed it buried in there somewhere), and these will directly scale the inferred timing of population size changes. The message that divergence is old is likely right – if gene flow occurs but isn't being captured by the analysis then divergence times should be *even older*, but if divergence time estimation is the goal then a demographic inference method explicitly incorporating gene flow and returning estimates of split times should be employed, and the empirical rates should be better justified. I'd suggest demographic modeling of the joint site frequency spectrum in dadi or moments. In addition, the end of this section regarding the possible speed of evolution of variation in migratory traits should acknowledge that much of that literature is based on the documented contemporary evolution of the NW migration and not on biogeographic reconstruction of island colonizations.

Thank you for this suggestion. In the revised version of our manuscript, we complemented our demographic inference with cross-coalescent rate analyses. The approach that we used was introduced by the MSMC model (Schiffels and Durbin, 2011) and extended in MSMC2 (Malaspinas et al., 2012). Comparing the rate of coalescence between pairs of individuals taken in two populations to the rate of coalescence within each population allows dating the differentiation of two populations. [Note: I would not copy the figure, just refer to the original paper.] [Eventually write:] This approach is particularly well-suited to our dataset as it does not require extensive sample sizes (as for instance needed by SFS estimation) and make use of complete genome data as it intrinsically accounts for linkage disequilibrium. In addition to the dating of population differentiation, cross-coalescence rates can be used to infer the temporal pattern of migration rate between populations (https://www.biorxiv.org/content/10.1101/585265v1.full). As our aim is to date population divergence, we decided to restrict our analyses to the differentiation timing and did not compute estimates of migration rates.

Subsection “The transition to residency may be controlled by regulatory elements” paragraph two: It was hard to tell what in this paragraph referred to results from this study, vs results from previous studies. Are Clock/Npas2/bmal1 in an outlier region here? Or is it just that there is one bHLH motif found in an outlier region? If the latter, I'd suggest cutting the second half of this section as the evidence of any specific regulatory element being involved seems quite weak.

We have made this clearer and while speculative we think it is important to include as the motif that corresponds with the literature search conducted by Ruegg et al., 2014, is located on Super-Scaffold_99 which shows parallel signatures of selection related to the transition from migration to residency in both continent and island birds.

Subsection “Examining genes associated with the transition to residency” paragraph one: Needs citations.

We have added references.

Subsection “Considerable variance in migratory distance is controlled by genetic variation”: This analysis did not seem particularly compelling, and (given that migratory distance is actually a continuous trait) the binning of distance as an ordinal trait seems likely to offer little power to identify causal alleles in a small cohort like this. It also isn't clear to me why a new method is used here that wasn't used in the previous association/selection analyses. I suggest cutting this section.

Migration distance is a continuous variable in theory but we do not have this kind of detail for our populations. We don’t know the exact distances an average bird from each population migrates. We did not want to lose the continuous nature of this variable entirely though so we chose to include it as an ordinal variable: short (1), medium (2), and long (3) distance migrants. We did not want to lose this information by running this analysis through hapFLK where it would be considered a categorical variable. GEMMA allows you to code your phenotype as ordinal. We have made this clear in our manuscript and think this analysis is integrated well enough as it is clear in our statement of objectives we will be studying the genetic basis of all three traits and this is restated at the start of the localized analyses as well.

Subsection “Selection on shared genetic variation could facilitate rapid changes in migration”: I like this question quite a bit and think there are some cool results here, but I found this section hard to follow. I think it needs a rewrite, and the methods should be better justified. I suggest starting with a paragraph laying out expectations for what you expect given selection from novel vs standing variation (see Barret and Schluter, 2008, citation 67), and using only analyses that have been previously proposed and shown through either theory or simulation to distinguish these processes.

We appreciate the reviewers comment. Barrett and Schluter, 2008, make three suggestions for showing selection acts on standing genetic variation. We go through each of these below and note that the authors themselves stated that “none of [their] approaches [are] infallible” and they “identify possible difficulties with each”.

1) Barrett and Schluter, 2008, suggest looking for evidence of soft sweeps around regions under selection using different population parameters. This analysis relies on the assumption that nearby neutral alleles will sweep with beneficial allele and is largely descriptive. For example, recombination around an allele that has been segregating in an ancestral population for some time will break up associations with nearby neutral sites. This should result in a narrow, shallow reduction in polymorphism compared to hard sweeps. Barrett and Schluter, 2008, do not provide any guidance on how to quantify how “narrow” and “shallow” regions have to be under this scenario. In addition, by their nature these signatures are more difficult to detect (narrow and shallow signatures will stand out less than broad, deep patterns). Finally, depending on how long the allele has been in the population any signature of a sweep could be eliminated completely by recombination. The latter point was noted by Barrett and Schluter, 2008. Given the difficulties associated with this method we have chosen not to employ it but we do note in the text that the peaks we identified are quite narrow.

2) Barrett and Schluter, 2008, also suggest finding evidence the allele under selection still occurs as standing variation in the ancestral population. This suggestion obviously requires us to know and still have an ancestral population to sample. This is easier to do in some systems (e.g. Colosimo et al., 2005 focus on an allele associated with armour in freshwater sticklebacks and are able to sample that marine population for this allele); we have done our best, noting that the migratory phenotype is likely ancestral in blackcaps (based on previous work to ours) and most of the haplotypes under selection are present in migratory populations (e.g. Figure 4). If these are the ancestral populations (suggested by previous authors including Perez-Tris et al., 2004 and Voelker and Light, 2011) our data on this topic support the use of standing genetic variation. A similar argument was made by Zhan et al. (2014) focusing on monarch butterflies where multiple losses of migratory behaviour involved selection at the same haplotype that is still present in the migratory population.

3) The last suggestion by Barrett and Schluter, 2008, is to use a phylogenetic study to determine the age of genomic regions under selection. Again their example comes from the Colosimo et al., 2005 paper on sticklebacks. These authors compared estimates of neutral sequence divergence between populations that differ in their armour phenotype to splits of known age within the same system to show the allele under selection is quite old. Unfortunately we do not have any closely related population pairs of known age to calibrate estimates of divergence in blackcaps but we have estimated neutral sequence divergence (the number of synonymous segregating sites) in all migratory populations and do not find a difference. Please note, this suggestion by Barrett and Schluter, 2008, may not be very reliable as the regions we’re looking at have undergone selective sweeps and patterns of neutral divergence may be obscured by the effects of selective sweeps on polymorphisms. This would make it difficult to use diversity to age divergence between haplotypes and was not noted by Barrett and Schluter, 2008.

We have also compared sequence divergence between all haplotypes and our ancestral sequence (derived using a comparison with two closely related species, the garden warbler (migratory) and hill babbler (resident)]).

We have also added two new analyses. First, we compare a phylogeny built using all data (ML tree) to one built using data from the genomic region under selection on Super-Scaffold 73 only (Figure 6). The genome-wide phylogeny is largely unresolved and has the garden warbler and hill babbler as a sister clade to blackcaps (as noted in many other phylogenetic studies on the *Sylvia* clade). By contrast, the phylogeny built using data from Super-Scaffold 73 suggests garden warblers are more closely related to blackcaps and medium distance NW birds diverged first, with the remaining blackcaps forming a polytomy as in the genome-wide tree. Garden warblers are migratory and the placement of NW birds at the base of the phylogeny suggests the haplotype in this population old and had been in the population before it underwent selection in the NW phenotype.

Our second new analysis is related to a pattern documented by Przeworski and Coop, 2011. These authors showed that SFS for regions experiencing selection from standing genetic variation are often more variable then those from de novo mutations and have more alleles at intermediate frequencies (see text for additional details). We note exactly this pattern in our study (Figure 6). Note this analysis involves resampling and a comparison of SFS from our data and the resamples. It would be ideal to compare the distribution of our observed data to resamples but at this time we are not aware of any way to compare distributions with these patterns (e.g. high frequencies at one end.).

Subsection “GWAS.”: What specifically was the factor used to represent population structure here? A matrix of pairwise genetic distance over the whole genome? Please provide a little more detail on your implementation of this method.

It is a kinship matrix generated using genome-wide SNP data.

Figure 1C: These colors seem to partially but not entirely match the map, which is confusing (especially when referring back to this figure for color references). I'd use different colors than in the map. Greyscale would work ok with K=3.

They are the exact same colours and we feel are instructive, helping the reader quickly see that there is admixture in both resident groups from the migrants. Accordingly, which we appreciate this comment we have chosen to stick with our original colour scheme.

Figure 5 legend: the letters seem to be off here – please check that e and f are correctly referenced, and discuss panels in order (a-e).

Thank you for noticing this. We have made the change.

[Editors’ note: what follows is the authors’ response to the second round of review.]

Reviewer #1:I find the resubmitted manuscript to be a substantial improvement over the initial submission. The authors are to be commended for their thorough revision and for incorporating reviewer comments.The only major comment from my previous review that I feel is not clarified in the revision is the way the three different migratory phenotypes are combined in the demographic analyses and the initial hapFLK/PBS/nSL analyses that include all populations (point #7 from my review). […] Some clarification of exactly which populations and phenotypes are being compared in the analyses, throughout this this section is needed, as well as justification for why medium- and long- distance migrants are combined for some analyses but not others, and clear reporting of sample sizes for each comparison in the main text (please remove from legend in Figure 3).

Thank you for this comment. There appear to be two problems here: (1) why were medium and long distance migrants combined for the demographic analysis (and in Figure 5A) and (2) what samples were used for analyses on the genetic basis of different migratory traits. We will address these concerns in turn below.

1) Why were medium and long distance migrants combined for demographic analyses (and in Figure 5A)?

Each one of the orientations (NW, SW and SE) were plotted individually in Figure 5A. They show similar trends making this difficult to see and our figure legend lacked clarity. We have improved the legend for Figure 5A stating that medium distance migrants were not combined for this plot.

We combined medium and long distance migrants for the demographic analyses because they exhibited very little population structure or differentiation (Figure 1) and thus we assumed they would exhibit similar demographic trajectories. We have confirmed this assumption now, running demographic analyses separately for each phenotype. The trajectories are indistinguishable and can now be found in Figure 2—figure supplement 4. We also reran cross-coalescent analyses and no population splits were observed (relative CCR stay close to 1 all the time; Figure 3—figure supplement 1). This information is in the main text as well.

2) What samples were used for analyses on the genetic basis of different migratory traits?

Analyses of population structure and demography included the full dataset. Subsets of the dataset were used in analyses focused on the genetics of migratory traits. We have included three new columns in Supplementary file 4 showing the exact birds used in each subset and – as requested by the reviewer – we have moved information on sample sizes from figure captions to the main text.

We have also removed the panel in Figure 3—figure supplement 1 showing estimates of PBS for residents as this is already shown in the main text (Figure 3A). Six groups (residents [shown in Figure 3A], short distance SW, medium distance NW, SW, SE and long distance SE birds were used in this analysis.